# Identification of an inhibitory neuron subtype, the L-stellate cell of the cochlear nucleus

Tenzin Ngodup[1], Gabriel E Romero[2], Laurence O Trussell[1]*

[1]Oregon Hearing Research Center and Vollum Institute, Oregon Health and Science University, Portland, United States; [2]Physiology and Pharmacology Graduate Program, Oregon Health and Science University, Portland, United States

**Abstract** Auditory processing depends upon inhibitory signaling by interneurons, even at its earliest stages in the ventral cochlear nucleus (VCN). Remarkably, to date only a single subtype of inhibitory neuron has been documented in the VCN, a projection neuron termed the D-stellate cell. With the use of a transgenic mouse line, optical clearing, and imaging techniques, combined with electrophysiological tools, we revealed a population of glycinergic cells in the VCN distinct from the D-stellate cell. These multipolar glycinergic cells were smaller in soma size and dendritic area, but over ten-fold more numerous than D-stellate cells. They were activated by auditory nerve and T-stellate cells, and made local inhibitory synaptic contacts on principal cells of the VCN. Given their abundance, combined with their narrow dendritic fields and axonal projections, it is likely that these neurons, here termed L-stellate cells, play a significant role in frequency-specific processing of acoustic signals.

## Introduction

In the ventral cochlear nucleus (VCN), auditory nerve (AN) afferents make synapses onto multiple subtypes of excitatory neurons, setting up parallel streams of processing that are used by higher centers for processing auditory cues for sound intensity, frequency, and localization. This early-level activity of excitatory neurons is sculpted and stabilized by inhibitory neurons, which use the transmitter glycine (*Altschuler et al., 1986*; *Wenthold et al., 1986*; *Wu and Oertel, 1986*; *Kolston et al., 1992*; *Ferragamo et al., 1998a*; *Chanda and Xu-Friedman, 2010*; *Xie and Manis, 2013b*; *Xie and Manis, 2014*). For example, glycinergic inputs onto excitatory bushy and T-stellate (i.e., Trapezoid body projecting) cells enhance acoustic tuning properties and temporal precision, and stabilize neuronal firing responses to acoustic inputs (*Caspary et al., 1994*; *Kopp-Scheinpflug et al., 2002*; *Gai and Carney, 2008*; *Keine and Rübsamen, 2015*). However, the sources of these inhibitory glycinergic inputs onto the principal cells in the VCN are not well understood.

Within the VCN, only a single glycinergic inhibitory cell type, called the D-stellate or radiate multipolar cell, has been described (*Oertel, 1983*; *Sento and Ryugo, 1989*; *Oertel et al., 1990*; *Doucet and Ryugo, 1997*; *Oertel et al., 2011*; *Xie and Manis, 2014*). D-stellate (i.e., Dorsal projecting) cells are a small population of neurons (*Doucet and Ryugo, 1997*; *Doucet et al., 1999*) that send their axons to the ipsilateral dorsal cochlear nucleus (DCN) (*Smith and Rhode, 1989*; *Oertel et al., 2011*; *Campagnola and Manis, 2014*) but also to sites as distant as the contralateral cochlear nucleus (CN) (*Schofield and Cant, 1996b*; *Needham and Paolini, 2003*; *Smith et al., 2005*). Importantly, D-stellate cells provide broadband inhibition to their targets, as their large dendritic arbors receive input from a broad frequency spectrum of auditory nerve fibers (ANFs) (*Smith and Rhode, 1989*; *Oertel et al., 1990*). Another source of inhibition in VCN is a projection neuron originating in the DCN, the tuberculoventral cells (also called vertical cells) (*Wickesberg and*

*For correspondence:
trussell@ohsu.edu

Competing interests: The authors declare that no competing interests exist.

*Oertel, 1988*; *Wickesberg et al., 1991*; *Xie and Manis, 2013a*; *Campagnola and Manis, 2014*). Recent studies suggest that D-stellate and tuberculoventral cells do not fully account for the inhibition observed in VCN principal cells *in vivo* (*Keine and Rübsamen, 2015*) and *in vitro* (*Campagnola and Manis, 2014*). Indeed, no local inhibitory interneuron has ever been described for VCN, which is unlike most known brain regions. Early anatomical studies suggested the presence of short axon cells in the VCN (*Lorente de Nó, 1981*). Furthermore, previous histological studies also reported the presence of small inhibitory neurons (glycinergic or possibly GABAergic) that are different from the D-stellate cells, but the identity of such neurons is unknown (*Wenthold, 1987*; *Kolston et al., 1992*; *Doucet and Ryugo, 1997*; *Gleich and Vater, 1998*; *Doucet et al., 1999*).

Here we comprehensively examined the diversity of inhibitory neurons in the VCN using a well-characterized transgenic mouse line, GlyT2-GFP, (*Zeilhofer et al., 2005*; *Kuo et al., 2012*; *Moore and Trussell, 2017*) which labels virtually all glycinergic neurons in the CN. With the use of this transgenic mouse, as well as optical tissue clearing, whole CN super-resolution microscopy, electrophysiological, and morphological tools, we discovered a large population of inhibitory glycinergic cell types distinct from the D-stellate cell. The novel glycinergic neurons, which have a narrower dendritic field than D-stellate cells, form the vast majority of inhibitory neurons in the VCN. We show that these cells, which we have termed L-stellate cells, receive monosynaptic ANF inputs and polysynaptic inputs from axon collaterals of T-stellate cells, and in turn locally inhibit bushy and T-stellate cells in the VCN. Thus, the VCN has a rich diversity of glycinergic interneurons to provide maximum flexibility to control the excitability of projection neurons in the VCN.

## Results

### Cell counts and soma size

We used a well-characterized GlyT2-GFP transgenic mouse (*Zeilhofer et al., 2005*; *Albrecht et al., 2014*; *Moore and Trussell, 2017*) in order to study the prevalence of glycinergic cells in the VCN. The neuronal glycine transporter, GlyT2, is a reliable marker of glycinergic cells, and GFP is selectively expressed in >90% of glycinergic cells in the CN. To visualize all the glycinergic cells, we optically cleared whole CN (450–500 µm) using CUBIC-mount (*Lee et al., 2016*). Next, we imaged the entire CN, lateral to medial (*Figure 1*). *Figure 1A* shows a series of 50 µm thick image stacks of CN, lateral to medial. Not surprisingly, we observed a dense population of glycinergic cells in the DCN as described in previous studies (*Oertel and Wu, 1989*; *Zhang and Oertel, 1993b*; *Zhang and Oertel, 1993a*; *Kuo et al., 2012*; *Apostolides and Trussell, 2014*). Also apparent were thick tracts of glycinergic fibers that entered the dorsal part of VCN, presumably projections of the glycinergic tuberculoventral neurons (vertical cells) in the DCN (*Figure 1A*). There was an obvious lack of glycinergic cells in the octopus cell region of the posterior VCN, consistent with previous studies (*Wickesberg and Oertel, 1988*; *Wickesberg et al., 1991*). However, we found a large population of glycinergic cells distributed across the rest of the VCN. To obtain a global view of their distribution, the images were stitched and combined to create a 3D image of the entire CN (*Figure 1Bi, ii*). The high density of glycinergic cells throughout VCN was surprising, because D-stellate cells are thought to be sparse, and the only other known glycinergic cells are Golgi cells, which are present mainly in the granule cell layer overlying VCN (*Ferragamo et al., 1998b*; *Irie et al., 2006*; *Yaeger and Trussell, 2015*).

We restricted our analysis to neurons only in VCN by masking the area outside the VCN and then quantifying the glycinergic cell count using semi-automated 'spot function' in Imaris software (*Figure 2A,B*). This counting procedure yielded a total of 2706 ± 107 glycinergic neurons in the VCN (n = 4 VCNs, three mice). Next, we quantified the soma volumes of all glycinergic neurons using the 'surface function' in Imaris (*Figure 2C,D*), and found that the soma volume distribution was positively skewed (*Figure 2E*), such that the vast majority of the glycinergic cells had small somas, and a minority had large somas (*Figure 2E*, inset).

To verify that GFP expressing neurons in the VCN from GlyT2-GFP mice were indeed glycinergic, immunohistochemical staining against glycine was performed on VCN sections (*Figure 2F*). The proportion of GFP expressing cells also positive for glycine was 93.44 ± 1.54%, whereas the proportion of glycine-positive cells also expressing GFP was 99.5% (n = 442 cells, n = 3 VCNs, three mice).

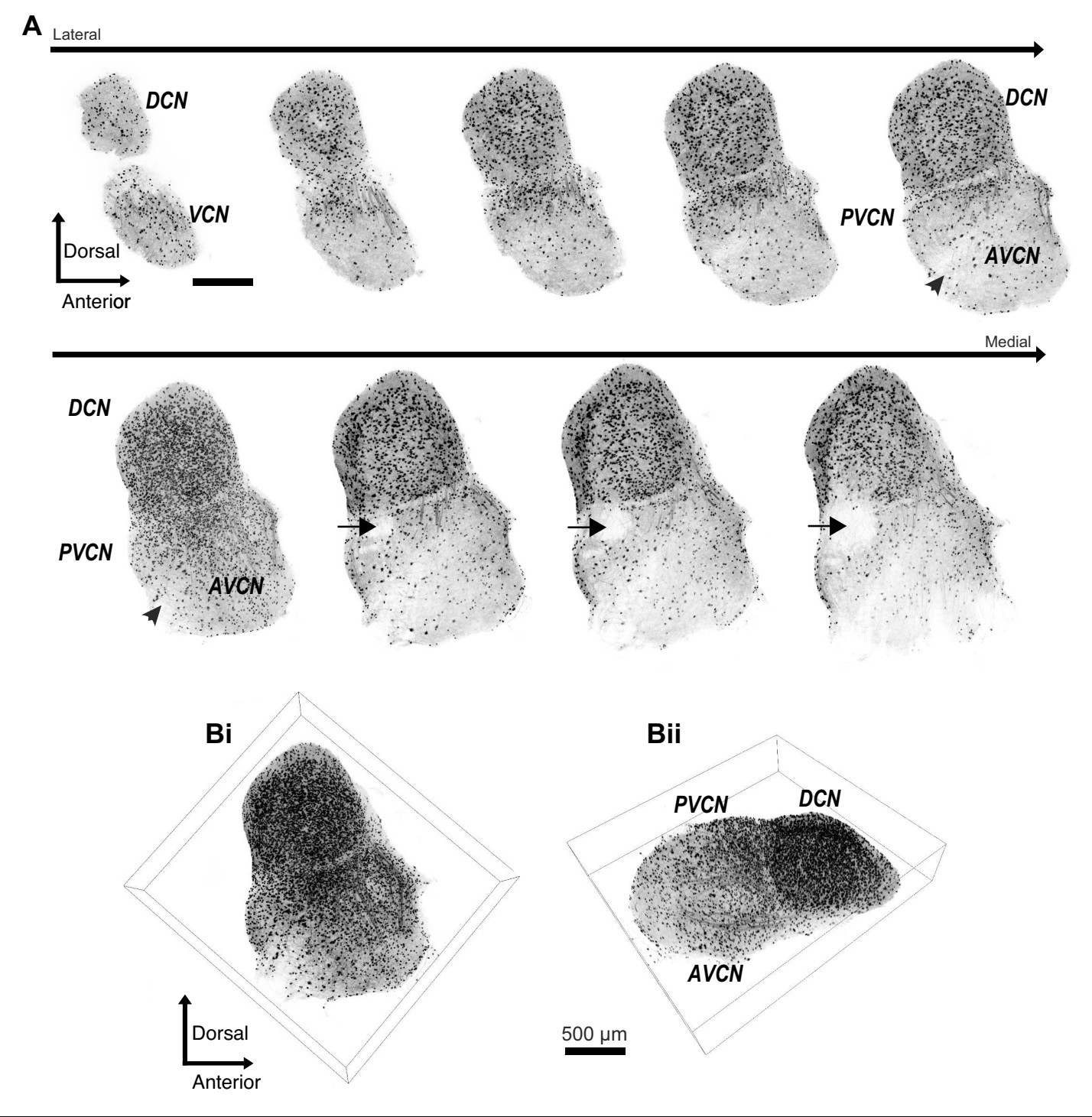

**Figure 1.** Diversity of glycinergic cell population in the CN. (**A**) Entire CN was optically cleared using CUBIC-mount and imaged with a super-resolution confocal microscope. Examples of 50 μm thick image stacks of CN from lateral to medial sides. DCN has a dense population of glycinergic cells. Octopus cell region in the PVCN (arrow) shows a lack of glycinergic cells. Arrowhead: auditory nerve root. (**Bi, ii**) 3D images of the entire CN with different viewpoints show dense population of glycinergic cells not only the DCN but also in the VCN. Source data is available at https://doi.org/10. 5061/dryad.69p8cz8xp. The zip archive, Figure_1_source_Data1_CN_series.tif.zip, contains CN images used for the quantitative analyses shown in *Figures 1* and *2*. Images were collected at 5 μm steps using Zeiss LSM880 with Airyscan super resolution microscope with 25× objective.

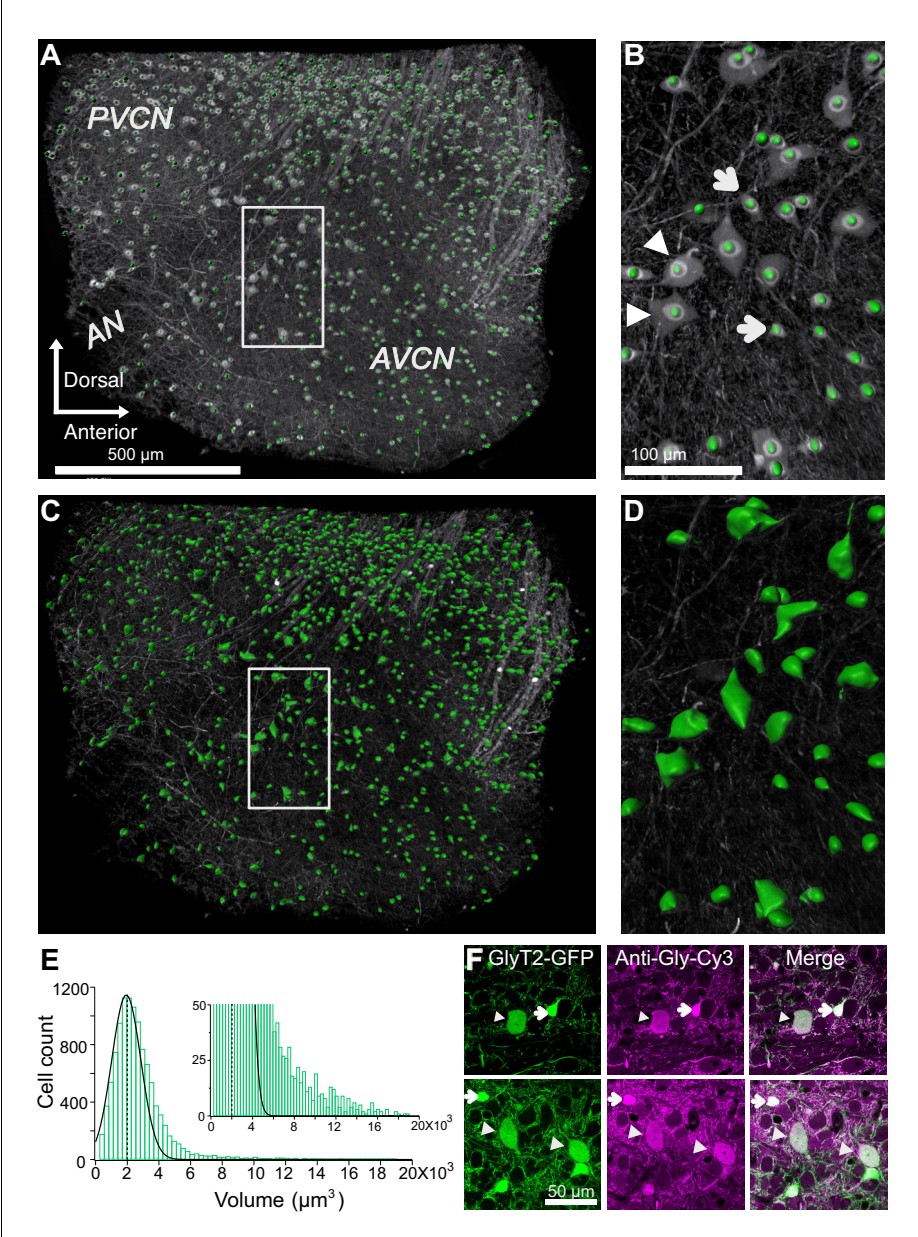

**Figure 2.** Quantification of glycinergic cells in the VCN. Maximum image projection of a 150 µm thick VCN. Cell count and soma size quantification were restricted to glycinergic cells in the VCN. (**A–B**) Cell counts were quantified using 'spot function' in Imaris. Each green dot is counted as one cell. (**B**) There is an anatomically distinct glycinergic cell population in the VCN. Example image shows a mix of large (arrowheads) and small cells (arrows). (**C–D**) Soma volumes were measured with surface rendering program in Imaris. (**E**) Soma size distribution was positively skewed. Dashed line shows the average soma size. The distribution was fitted by a Gaussian curve. Majority of the glycinergic cells had small soma size, and minority have large soma size (inset). (**F**) Two representative images of VCN showing large (arrowheads) and small (arrow) glycinergic cells colabeled for glycine. Source data is available at https://doi.org/10.5061/dryad.69p8cz8xp. The spreadsheet Figure_2-Source_Data_1_cell_volume1.xlsx contains all the raw volume data for the four CNs used for the quantitative analyses shown in *Figure 2*. The individual files and their data are present in separate sheets.

Interestingly, RNA probes against the *Slc6a5* gene (GlyT2) also clearly reveal both large and small glycinergic cells types in VCN (https://mouse.brain-map.org/experiment/show/69874024).

To study the existence of multiple glycinergic sources, we took an intersectional approach to test whether the population of glycinergic neurons in VCN consists of molecularly distinct glycinergic cell types. We used a somatostatin-Cre (SST-Cre) mouse line that has been used to identify and study somatostatin-containing neurons in the brain. An SST-Cre::Ai9 mouse line expresses tdTomato in a variety of neurons in the VCN. This mouse line was crossed to the GlyT2-GFP mouse line. In the

resulting cross, large GFP-positive cells were clearly positive for tdTomato; given their size, these are candidate D-stellate cells, and this inference was confirmed by experiments described below on commissural projections.

However, closer examination revealed that the majority of the GFP positive cells were tdTomato-negative and thus formed a size class distinct from the double-labeled cells presumed to be D-stellate cells. GFP and tdTomato fluorescence was amplified with antibodies to ensure the visualization of cells with lower expression (n = 3 VCN, two mice) (*Figure 3A,B*). There was a strong correlation between GFP and tdTomato expression in presumptive D-stellate cells (*Figure 3B*, top row). By contrast, smaller cells that were GFP positive were clearly tdTomato negative (*Figure 3B*, bottom row). Thus, the small glycinergic cells were molecularly distinct from the larger D-stellate cells in that only the latter expressed Cre in this mouse line.

This conclusion was further probed by examining the distributions of small glycinergic cells and D-stellate cells in the VCN. CNs from SST-tdTomato::GlyT2-GFP mice were optically cleared using the CUBIC-mount (*Figure 4Ai–iii*). We then separately measured the soma volumes of glycinergic cells expressing GFP only and D-stellate cells expressing both GFP and tdTomato using the surface function in Imaris (*Figure 4Bi–v*). We found a broad distribution of soma size of double-labeled, D-stellate cells. This suggest that soma size alone is not a predictive indicator of cell class. However, we found that on average double-labeled cells were significantly larger than GFP-only cells (average soma volume, GFP only, $1826.6 \pm 12.9$ $\mu m^3$ versus D-stellate cells, $4289.5 \pm 98.3$ $\mu m^3$, n = 2 VCNs, two mice). However, numerically, the larger cell types composed only about 12% of all glycinergic neurons in VCN (GFP only, n = $3250 \pm 55$ versus D-stellate cells, n = $380 \pm 9$) (*Figure 4C*, inset). Our data thus indicate that D-stellate cell types and small glycinergic cell types are molecularly distinct populations of glycinergic interneurons in the VCN, and also shows that the small cells compose the vast majority of glycinergic neurons in the VCN.

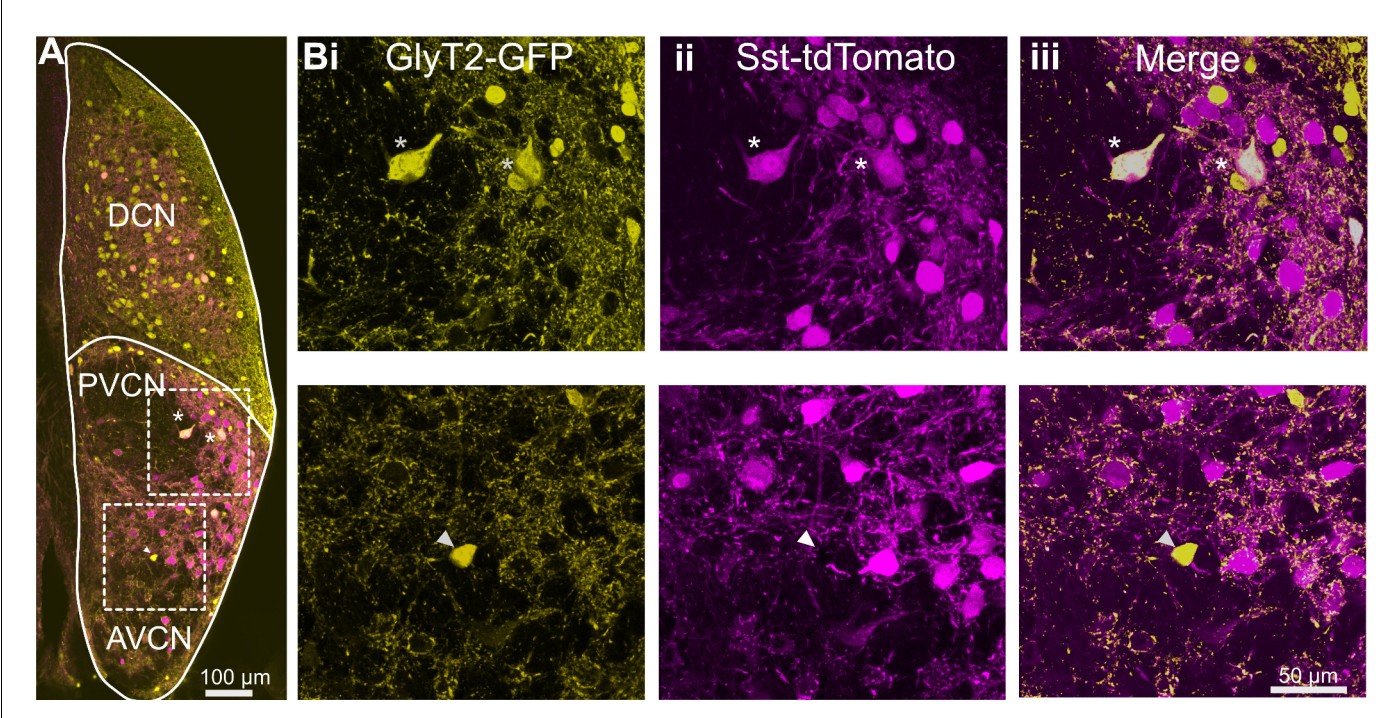

**Figure 3.** D-stellate cells are molecularly distinct from the small glycinergic cells. (**A**) Coronal section of a CN from an SST-tdTomato::GlyT2-GFP mouse. (**Bi-iii**) D-stellate cells show strong colocalization of tdTomato and GFP (inset and top row, asterisk) whereas small glycinergic cells were tdTomato negative and GFP positive (bottom row).

## Axon-dendritic arbors of cell types

To further confirm that the small glycinergic cells are a distinct population, we performed individual cell fills by patch clamping GFP-positive neurons with pipettes containing biocytin in the intracellular solution, and reconstructing the filled cells using Neurolucida (*Figure 5*). Large cells exhibited typical radiate morphology which have been described previously for D-stellate cells in the VCN (*Oertel et al., 1990*; *Campagnola and Manis, 2014*; *Xie and Manis, 2014*) (n = 6) (*Figure 5A*), and henceforth will be referred to as D-stellate cells. Unlike D-stellate cells, the small glycinergic cells had restricted axonal and dendritic arbors (n = 17) (*Figure 5B*). The spread of the axonal-dendritic field was quantified by measuring the longest and shortest axis of the reconstructed structures. The small glycinergic cells had significantly restricted branched processes compared with D-stellate cells (longest axis: D-stellate cells, 619.42 ± 45.78 μm versus small cells, 271.43 ± 26.27 μm, p<0.001, t-test; shorter axis: D-stellate cells, 435.62 ± 1.82 μm, n = 6 cells versus small cells, 148.60 ± 12.03 μm, n = 17, p<0.001, t-test) (*Figure 5D*). We also measured the volume and surface area under the axonal-dendritic field assessed by its convex hull, and found that the field encompassed by D-stellate cell processes had a larger volume and surface area than that of small glycinergic cells (*Figure 5E and F*). Moreover, the small cells showed significantly higher axonal-dendritic arborization closer to their somata as compared with large D-stellate cells (Sholl analysis, p=0.006, K-S test) (*Figure 5F*). Our morphological data indicate that smaller glycinergic cells are anatomically distinct from D-stellate cell types and provide further evidence that small glycinergic cells are a distinct class of inhibitory neuron. Finally, we compared the morphology of small glycinergic cells with excitatory T-stellate (planar) cells, defined as non-GFP containing, dendritic neurons with chopper firing properties (*Figure 5C*). We found that small glycinergic cells have a similar spread of the axonal-dendritic field in the longest axis but significantly larger spread in the shortest axis (longest axis: T-stellate cells, 237.03 ± 16.67 μm versus small cells, 271.43 ± 26.27 μm, p=0.33, t-test; shorter axis: T-stellate cells, 79.73 ± 13.68 μm versus small cells, 148.60 ± 12.03 μm, p<0.005, t-test) (*Figure 5D*). This suggests that small glycinergic cells are less planar than T-stellate cells. We also found that small glycinergic cells had more profusely branched arbors near their soma whereas T-stellate cells exhibited more branching at the end of dendrites, which is consistent with the previous studies (*Rhode et al., 1983a*; *Oertel et al., 1990*; *Figure 5G*, Sholl analysis, p<0.05, K-S test). These data confirm that small glycinergic cells are morphologically a distinct class of neuron in the VCN.

## Intrinsic properties

Next, we studied the intrinsic electrical properties of the D-stellate and the small glycinergic cells in the VCN by targeting GFP positive cells from GlyT2-GFP mice for whole-cell recordings. D-stellate cells were identified based on their large soma size and dendrites, while the small cells had obviously smaller somata and less distinct dendrites. Representative responses to depolarizing and hyperpolarizing current injections are shown in *Figure 6A,B*. All cells showed sustained firing responses at weak-to-moderate depolarizing current steps; however, their responses to strong depolarizing current injections were more diverse (*Figure 6A–B*, top), particularly as regards the maintenance of spike amplitude during the response. Measurements of membrane properties, action potential shape, and firing properties were collected for all recorded glycinergic cells (*Table 1*). We used three parameters (action potential half-width, after-hyperpolarization [AHP] decay, and input resistance; see Materials and methods; *Figure 6C–E*, inset *Figure 6F*) to classify glycinergic cells into different clusters in order to probe for distinct inhibitory cell types (*Figure 6F,G*). The elbow method for estimating the optimal number of clusters (*Figure 6G*, inset) indicated the presence of two to three groupings of these cells. We then used K-means cluster analysis to divide the population of all the glycinergic cells into two groups (*Figure 6F*). Cluster analysis based on electrophysiology accurately separated 91.7% (11/12) of visually-identified D-stellate cells (red circles) from small cells (blue). However, the parameters used to classify the small cells were widely distributed which suggests there may be further subtypes among the small glycinergic cells. When we used a K-value of 3, the 'small' cells were further classified into two groups (*Figure 6G*, green versus blue circles). The high accuracy of the cluster method for classifying glycinergic cells suggest that small cell types may be solely identified based on the physiological membrane properties of the cells.

Many cells in the VCN, such as T-stellate cells (an excitatory principal neuron), exhibit 'chopping' responses to acoustic stimuli, such that the peristimulus time histograms of spiking show initial peaks

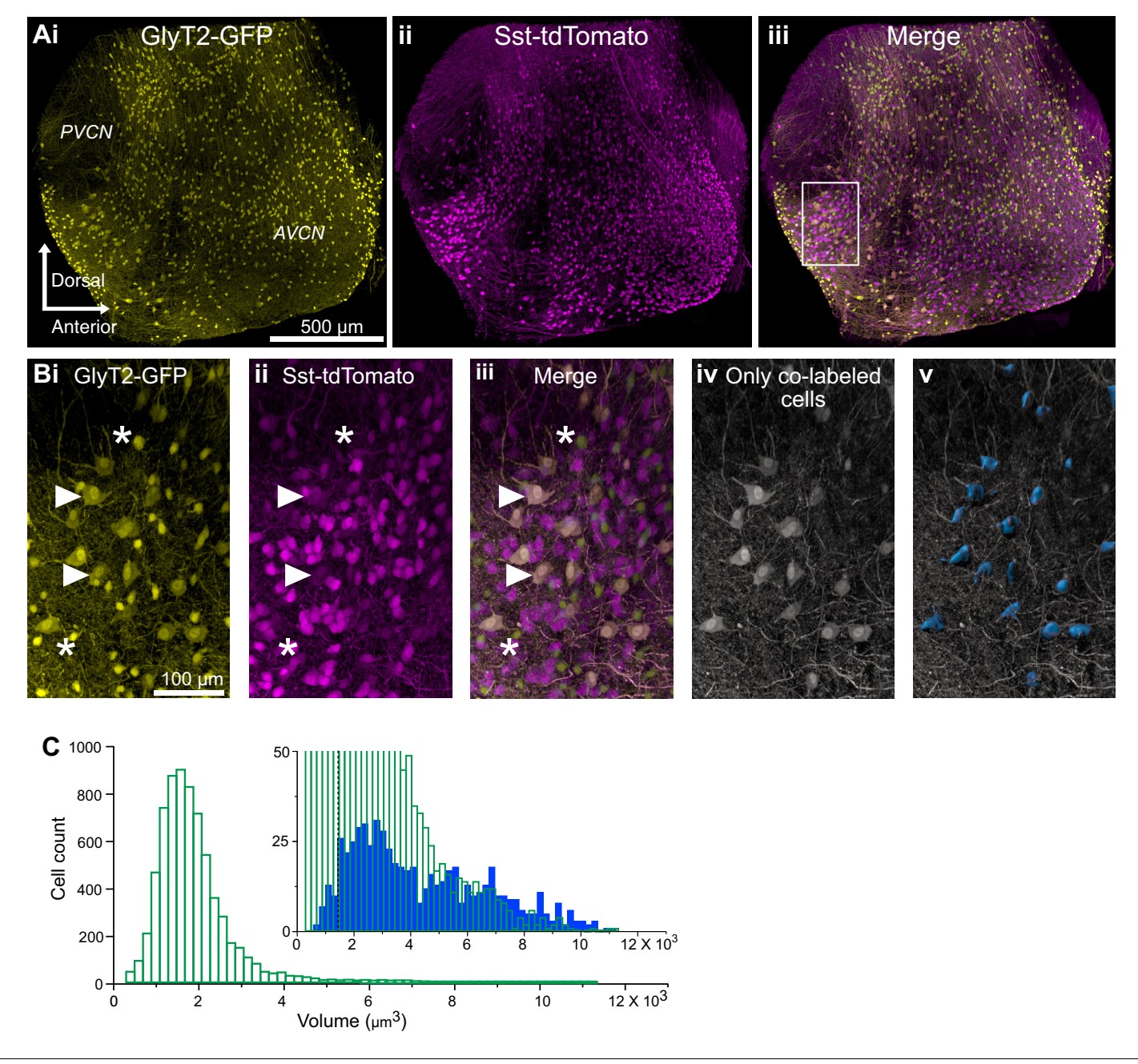

**Figure 4.** D-stellate cell types are molecularly distinct from the small glycinergic cells. (**A**) 150 μm thick image stacks of optically cleared VCN from SST-tdTomato::GlyT2-GFP mouse showing GFP (**i**), tdTomato expressing cells (**ii**) and merge (**iii**). (**B**) An example area from VCN (inset, **Aiii**) showing large GFP (**i**) and tdTomato (**ii**) expressing cells colocalize (**iii**). (**Biv-v**) Only the double-labeled cells are shown and surfaces were rendered to measure soma size. (**C**) Soma size distribution of all the glycinergic GFP positive neurons. Inset, soma volumes of double-labeled cells (blue) compared to all GFP positive cells (green). Double-labeled cells were significantly larger than GFP-only cells (average soma volume, GFP only, 1826.55 ± 12.93 μm$^3$ versus D-stellate cells, 4289.50 ± 98.29 μm$^3$, n = 2 VCNs, two mice). However, numerically, the larger cells types composed only about 12% of all glycinergic neurons in VCN (GFP only, n = 3250 ± 55 versus. D-stellate cells, n = 380 ± 9). Dashed line shows the average soma size. Source data is available at https://doi.org/10.5061/dryad.69p8cz8xp. The spreadsheet Figure_4-Source_Data_1_cell_volume_SSt.xlsx contains all the raw volume data for the 2 CNs used for the quantitative analyses shown in Figure 4.

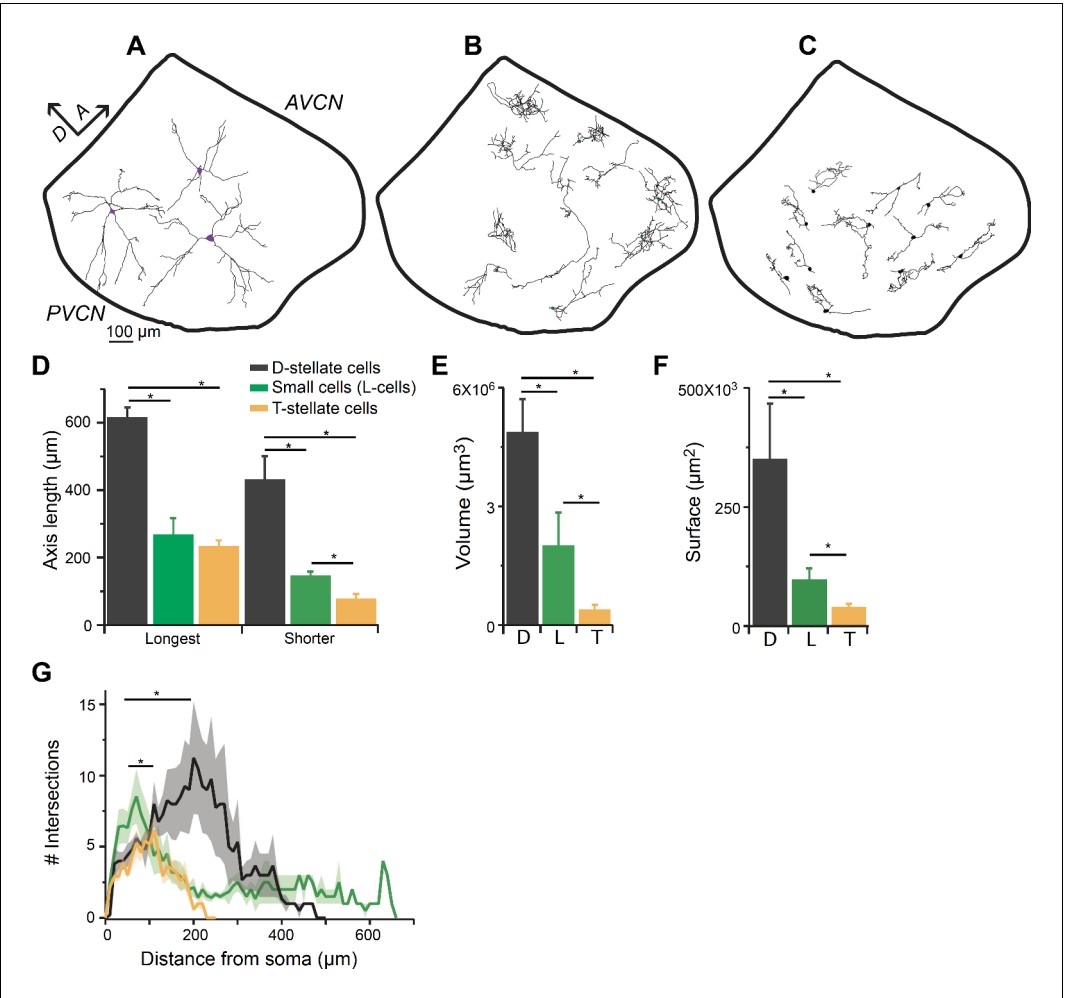

**Figure 5.** Smaller glycinergic cells are anatomically distinct from D- and T-stellate cells. Biocytin filled neurons were reconstructed using Neurolucida. (**A**) D-stellate cells (large cells) exhibited typical radiate morphology whereas (**B**) the smaller glycinergic cells show restricted axonal and dendritic arbors. (**C**) T-stellate cells show tuffed dendritic arbors and occupy regions parallel to the isofrequency bands. (**D**) Spread of axonal-dendritic arbors was quantified by measuring the longest and shortest axis of the reconstructed structures. Smaller glycinergic cells had significantly restricted axonal-dendritic process compared with D-stellate cells (longest axis: D-stellate cells, $619.42 \pm 45.78$ µm versus. small cells, $271.43 \pm 26.27$ µm, p<0.001, t-test; shorter axis: D-stellate cells, $435.62 \pm 1.82$ µm, n = 6 cells versus small cells, $148.60 \pm 12.03$ µm, n = 17, p<0.001, t-test). In comparison to T-stellate cells, small glycinergic cells had similar spread of the axonal-dendritic field in the longest axis but occupied a significantly larger region in the shortest axis longest axis: T-stellate cells, $237.03 \pm 16.67$ µm versus small cells, $271.43 \pm 26.27$ µm, p<0.33, t-test; shorter axis: T-stellate cells, $79.73 \pm 13.68$ µm versus small cells, $148.60 \pm 12.03$ µm, p<0.005, t-test. D- and T-stellate cells had significantly larger volume (volume: D-stellate cells, $4.8 \pm 1.2 \times 10^6$ µm$^3$ versus small cells, $2.02 \pm 0.69 \times 10^6$ µm$^3$, p<0.001, t-test) (**E**) and surface area (D-stellate cells, $20.91 \pm 1.07 \times 10^3$ µm$^2$ versus small cells, $9.83 \pm 2.71 \times 10^3$ µm$^2$, p<0.01, t-test) (**F**) compared with the small glycinergic cells. (**F**) Sholl analysis showing that small cells had significantly higher branching closer to soma compared with D- and T-stellate cells (Sholl analysis, p<0.05, K-S test).

of activity at regular intervals unrelated to the phase of the sound stimulus (*Rhode et al., 1983b*; *Young et al., 1988*; *Blackburn and Sachs, 1989*; *Smith and Rhode, 1989*; *Oertel et al., 2011*). No definitive *in vivo* recordings of small inhibitory cells in VCN have been reported. To test whether small cells have the capacity to generate such chopping responses, we recorded from the cells *in vitro* and applied repeated long depolarizing current steps (200 pA) (*Figure 7A*). The resulting peri-stimulus time histograms exhibited features that closely resembled *in vivo* 'sustained' chopper responses—with well-timed onset action potentials, and subsequent spikes which increasingly

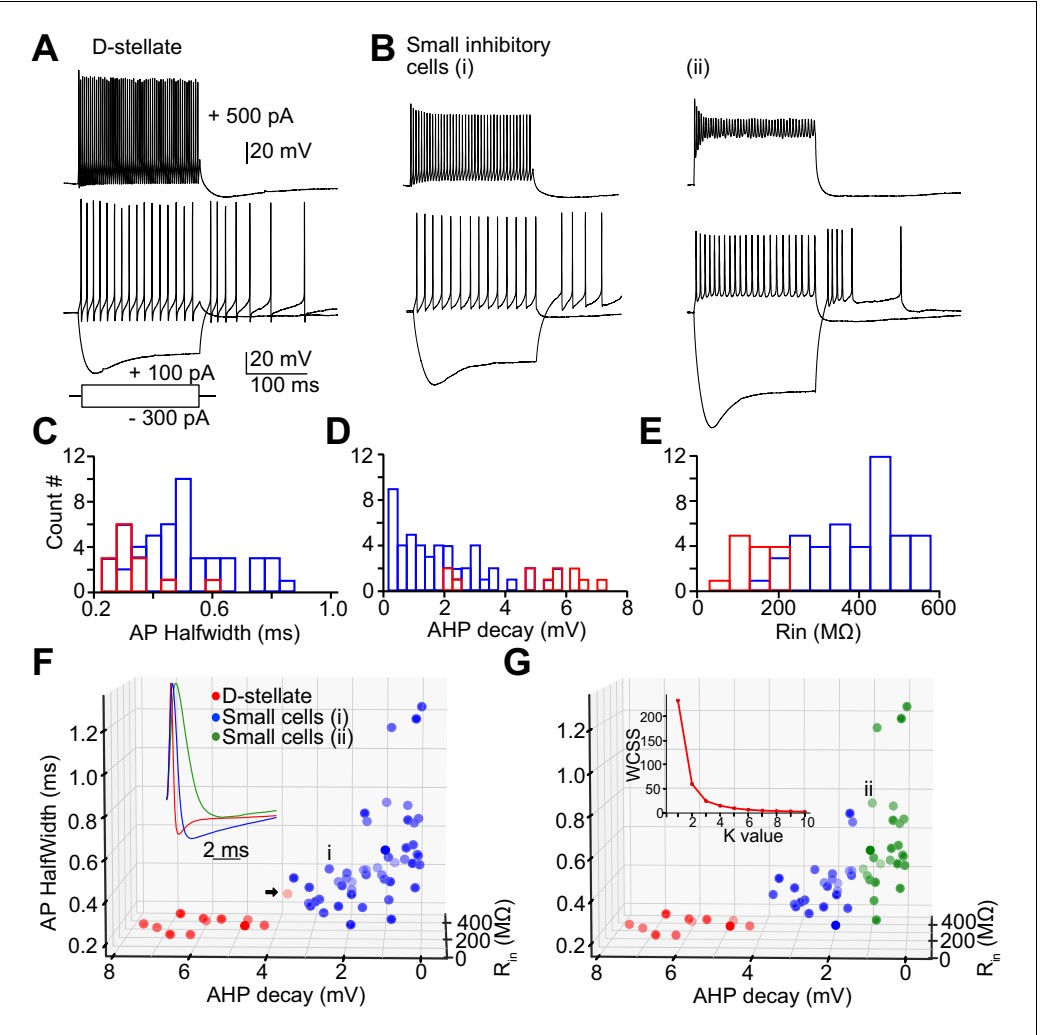

**Figure 6.** Heterogeneity of spiking properties of the small glycinergic cells. Representative examples of D-stellate cells (**A**) and small glycinergic cell types (**B**) responding to current injections (+500 pA, +100 pA, and −300 pA). All cells showed sustained firing responses to +100 pA current injections. Small cells exhibited diverse spike amplitude adaptation in responses to strong (+500 pA) current steps (top, **Bi-ii**). (**C–E**) Histogram of AP halfwidth, AHP decay, and $R_{in}$ (**F**) Classification of glycinergic cells based on the AP halfwidth, AHP decay, and $R_{in}$. The elbow method was used to calculate the optimal number of clusters (Panel 6G, inset [labels, Within cluster sum of squares (WCSS)]). Glycinergic cells are group into clusters based on K-means cluster analysis (K = 2) (red: D-stellate cells, blue: small cells, color gradient: front [dark] to back [light]). (**G**) K = 3 resulted in further classification of small cells into two sub clusters (blue and green). Arrow points to the D-stellate cell included in small cells when K = 3, i and ii represents small cells in blue and green clusters, respectively. Source data is available at https://doi.org/10.5061/dryad.69p8cz8xp. The spreadsheet Figure_6-Source_Data_1_intrinsic_properties.xlsx contains all the raw data used for the quantitative analyses shown in *Figure 6*.

The online version of this article includes the following figure supplement(s) for figure 6:

**Figure supplement 1.** Membrane properties of small glycinergic cells to carbachol sensitivity.

accumulated more temporal jitter throughout the duration of the current step, despite little to no adaptation in firing rate (*Figure 7B–D*, n = 26). This feature raises the possibility that these smaller neurons could give acoustic responses like those of principal cells, and possibly be mistaken for such cells in studies in which no other means for cell identification is used.

Table 1. Membrane properties of the glycinergic cells in the VCN.

$R_{in}$: Input resistance; AP height: action potential amplitude from threshold to peak; AP halfwidth: action potential width at half amplitude, measured between threshold and peak; AP AHP decay: voltage measured between peak of undershoot to Vm 1 ms later; AP AHP latency: action potential hyperpolarization latency; Rate of rise: rate of rise of action potential; Threshold: action potential threshold; Overshoot: peak of action potential from 0 mV; Undershoot: Peak of hyperpolarization from baseline; Firing rate: number of spikes/s. See the source data for *Figure 6* available at https://doi.org/10.5061/dryad.69p8cz8xp.

| Measure | D-stellate | Small cell |
|---|---|---|
| $R_{in}$ (MOhm) | 114.87 ± 12.77 | 305.54 ± 16.93 |
| AP height (mV) | 70.14 ± 2.59 | 63.92 ± 1.60 |
| AP halfwidth (ms) | 0.31 ± 0.02 | 0.55 ± 0.03 |
| AP AHP (mV) | 4.18 ± 0.62 | 1.55 ± 0.20 |
| AP AHP latency (ms) | 0.87 ± 0.10 | 1.81 ± 0.15 |
| Rate of rise (V/s) | 317.67 ± 17.85 | 288.06 ± 14.71 |
| Threshold (mV) | −45.57 ± 1.18 | −43.46 ± 0.49 |
| Overshoot (mV) | 24.57 ± 3.05 | 20.46 ± 1.59 |
| Undershoot (mV) | −17.79 ± 1.16 | −14.03 ± 0.59 |
| Firing rate (sp/s) | 213.52 ± 17.31 | 225.72 ± 14.94 |

## Synaptic connectivity

ANFs are the primary source of excitation to the VCN. However, collaterals of principal cells could in theory provide some local excitation (*Cao et al., 2019*). To test whether the small cells receive ANF inputs, we electrically stimulated the AN root in the presence of strychnine and gabazine to block inhibitory synaptic inputs, and recorded excitatory responses from small GFP-positive cells. Trains of shocks applied to the AN root readily led to postsynaptic spikes that exhibited a chopping pattern similar to that seen with current injection, although often with a period of 'extra' spikes after the end of stimulation (*Figure 7E–H*; 6 of 6 neurons). In voltage clamp, AN root stimulation evoked excitatory postsynaptic currents (EPSCs) that grew in size with the strength of the shocks (*Figure 8A*), indicating multiple ANF inputs per postsynaptic cell. Varying stimulus strength indicated that each cell receives 3–5 ANF inputs (*Figure 8A*). The average maximal amplitude of EPSCs evoked with AN root stimulation was 385.7 ± 49.9 pA (n = 22). The EPSCs also showed fast kinetics (τ = 0.80 ± 0.07 ms, half-width = 0.91 ± 0.08 ms, n = 22) with short and consistent synaptic delays of less than 1 ms, consistent with the monosynaptic transmission.

In many small cells (45%, 10/22), AN root stimulation evoked EPSCs with distinctly different latencies (*Figure 8B*). The presence of peaks with different latencies suggests that the small cells also receive polysynaptic excitation from neurons located within the brain slice. Specifically, the first EPSC always had a short synaptic latency consistent with a monosynaptic input (*Figure 8B*, black arrowheads), but this was often followed by EPSCs two or more ms later (*Figure 8B*, gray arrowheads). In the VCN, excitatory T-stellate cells have been shown to make local collaterals (*Oertel et al., 1990*; *Ferragamo et al., 1998a*) and can provide some local excitation (*Cao et al., 2019*) whereas bushy cells do not appear to make local collaterals (*Smith et al., 1991*; *Smith et al., 1993*). Moreover, *Fujino and Oertel, 2001* showed that T-stellate cells can be excited by the cholinergic agonist carbachol. We reasoned that if T-stellate cells are responsible for the delayed EPSCs, the frequency of those EPSCs should be increased if the T-stellate cells were made more excitable with carbachol. To test this idea, carbachol (10 µM) was bath applied while monitoring spontaneous EPSCs (sEPSCs) from small GFP-positive cells. We observed a significant increase in the amplitude and frequency of sEPSCs (*Figure 8C,D*), as expected if the T-stellate cells were firing spontaneously. In order to test the role of local collaterals in the delayed events, we recorded EPSCs in small GFP-positive cells evoked by AN root stimulation. A representative recording with 10 superimposed traces is shown in *Figure 8E*. In this cell, we observed a first EPSC with a short latency (<1 ms)

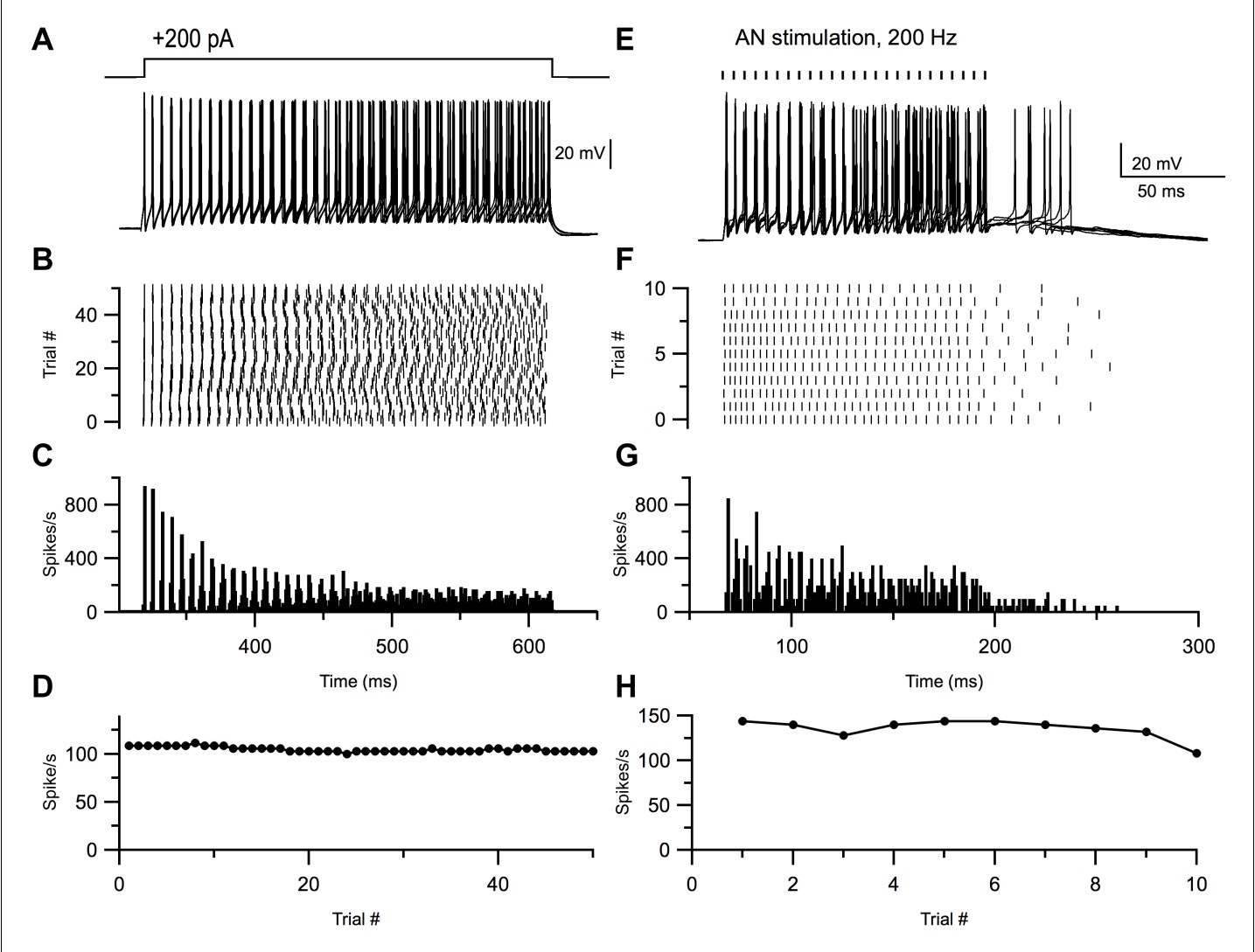

**Figure 7.** Small glycinergic cells show chopper responses. Representative example of a small glycinergic cell responding to +200 pA current injections (A) and electrical stimulation at 200 Hz (E). Raster (B, F), PSTH (C, G), and firing rate (D, H) of cell in A and E, respectively.

followed by events with longer latencies (>2 ms). Carbachol (10 µM) was then bath applied and EPSCs were recorded in response to the same AN root stimulation strength (*Figure 8F*). The latencies and numbers of EPSC were quantified and displayed as raster and PSTH plots (1-ms bin; *Figure 8G–J*). Across multiple cells (n = 5), there was no significant increase in the number of EPSCs with a latency <1 ms (percentage increase from control = 17.57 ± 0.11%). By contrast, we observed a significant increase in the number of events with longer latencies (percentage increase from control = 92.00 ± 0.13%). These results strongly suggest that the local collaterals from T-stellates are the source of polysynaptic excitatory inputs onto the small glycinergic cells.

Next, we examined the projections of the small GFP-positive neurons. D-stellate cells project into DCN (*Smith and Rhode, 1989*; *Oertel et al., 2011*), as well as to the contralateral CN (*Wenthold, 1987*; *Shore et al., 1992*; *Schofield and Cant, 1996b*; *Alibardi, 1998*; *Doucet et al., 1999*; *Needham and Paolini, 2003*; *Doucet and Ryugo, 2006*). In order to test whether the small GFP-positive cells also project to contralateral CN, we injected retrogradely-transported fluorescent latex beads (50–100 nL) unilaterally into CN of GlyT2-GFP mice (n = 3). Five days after the bead injection, we examined the coronal sections of CN ipsi- and contralateral to the injection (*Figure 9A,B*). Some GFP positive neurons were double-labeled with retrobeads in contralateral CN, however, virtually all of these had large somas, consistent with D-stellate cells (length of longest axis = 26.91 ± 3.92 µm,

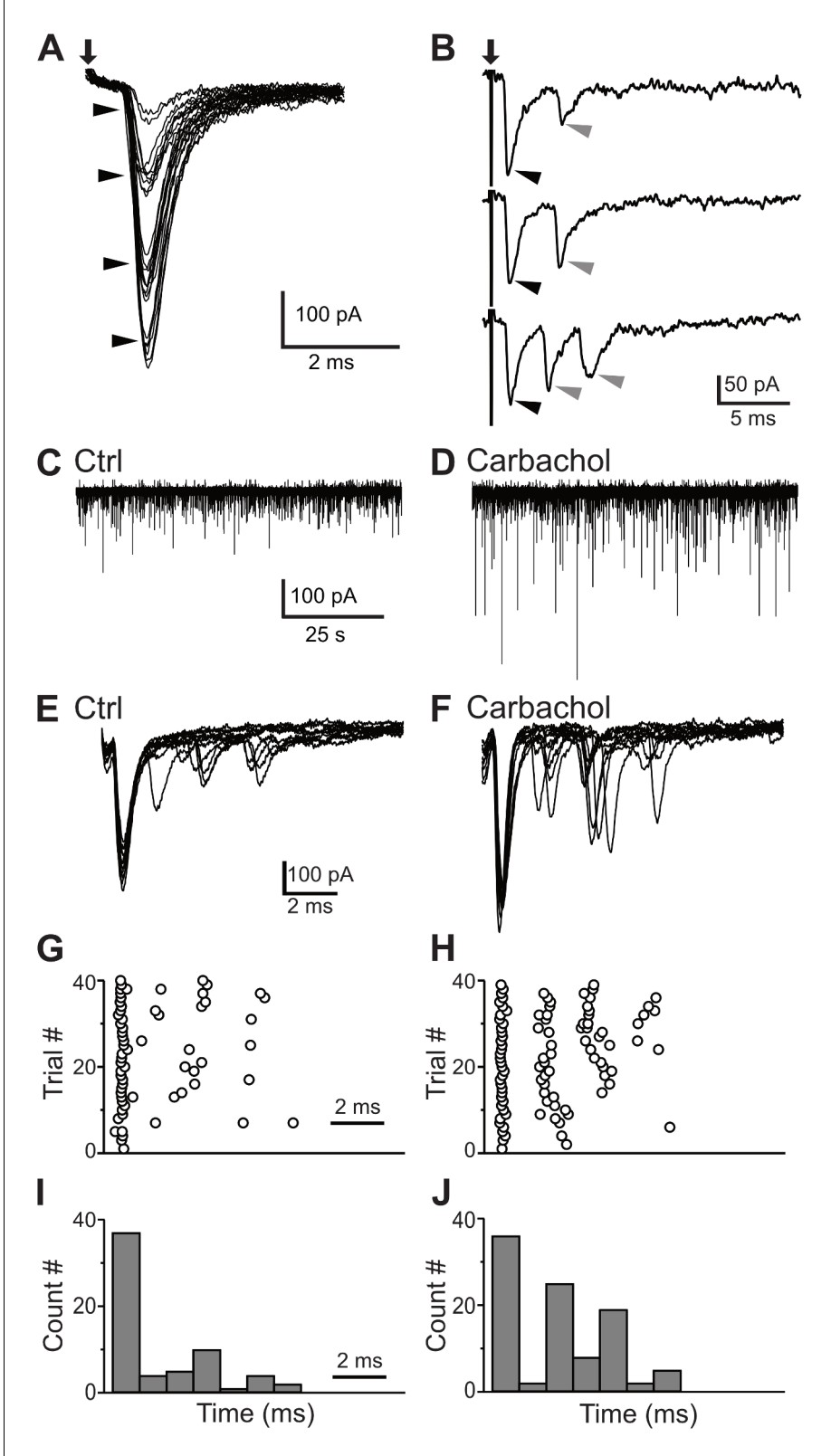

**Figure 8.** Synaptic inputs to the small glycinergic cells. (**A**) Stimulation of AN root evoked EPSCs with graded amplitudes. The EPSCs amplitude increased with stronger stimulus strength (arrowheads). (**B**) Representative example of disynaptic EPSCs with longer latency in response to AN root stimulation (gray arrowheads). (**C**) Example trace showing sEPSCs measured in small cells under control condition (**C**) and carbachol (**D**). The increased in sEPSCs frequency confirms T-stellate cells as disynaptic source of EPSCs. (**E**) Representative example of eEPSCs in response to AN root stimulation

*Figure 8 continued on next page*

*Figure 8 continued*

under control conditions (**E**), raster (**G**) and PSTH (**I**) of cell in **E**. (**F–H**) Carbachol application significantly increased the number of delayed EPSCs (short latency events < 1 ms, percentage increase = 17.57 ± 0.11%; longer latency events, percentage increase = 92.00 ± 0.13%).

n = 47, n = 3 CNs) (*Figure 9C*). Thus, few if any of the small GFP-positive cells project contralaterally, again indicating that they compose a separate cell class.

Our anatomical reconstruction data (*Figure 5*) of the small GFP-positive neurons in the VCN suggest these cells might provide local inhibition to principal cells in the VCN. To test this, we first tried paired-recordings between small cells and principal cells of the VCN, however, out of 40 paired-recordings, we found only a single successful pairing which a GFP-positive small cell generated IPSCs in a non-fluorescent VCN neuron. As the yield was so low we used a different approach. Interestingly, we have found that a subset (9/15) of the small GFP-positive cells can be activated by bath application of carbachol (10 µM; n = 15; *Figure 10A*). Carbachol sensitive and insensitive small cells did not differ in intrinsic properties (*Figure 6—figure supplement 1*). By contrast, the other major inhibitory sources to VCN, including the D-stellate (n = 6) (*Figure 10B*) and the tubercoluventral cells of the DCN (n = 5) (*Figure 10C*), failed to respond to carbachol. This lack of carbachol sensitivity in the D-stellate cells is consistent with observations reported by *Fujino and Oertel, 2001*. To examine whether the small cells provide inhibition to principal cells in the VCN, we enhanced their excitability by bath application of carbachol (10 µM) and recorded inhibitory postsynaptic currents (IPSCs) from bushy cells and T-stellate cells in the presence of excitatory synaptic blockers (10 µM NBQX, 5 µM MK-801). Carbachol application produced a clear increase in the frequency of spontaneous IPSCs (sIPSCs) in both bushy and T-stellate cells (*Figure 10D and E*) (inter-event interval, control = 280.4 ± 47.6 ms versus carbachol = 191.2 ± 29.0 ms, p<0.02, t-test, n = 18) (*Figure 10J*). The identity of the principal cells as both bushy and T-stellate cells were confirmed using their sIPSC kinetics. The decay time course of the sIPSCs (bushy cell, τ = 7.22 ± 1.40 ms, n = 9, T-stellate, τ = 2.28 ± 0.07 ms, n = 9) (*Figure 10F–I*), consistent with cell-specific kinetics reported previously (*Xie and Manis, 2013a*; *Xie and Manis, 2014*; *Lin and Xie, 2019*). Thus, the small GFP-positive, glycinergic cells likely act as local interneurons.

## Discussion

Subtypes of VCN stellate cells have been named according to the projections of their axons (D- for Dorsal, T-for Trapezoid bundle). Given their Local projections, we term the small, multipolar glycinergic cells 'L-stellate cells'. This study used transgenic animals, combined with molecular, morphological, and electrophysiological techniques to show that the L-stellate cells are molecularly, anatomically, and electrophysiologically distinct from the well-studied, D-stellate cell types. Moreover, L-stellate cells constitute the vast majority of glycinergic cells in the VCN. They receive monosynaptic AN inputs and feedforward excitatory inputs from local T-stellate cells, and their axons appear to inhibit principal excitatory cells, bushy cells, and T-stellate cells of the VCN (*Figure 11*). Therefore, the VCN has a distinct population of glycinergic interneurons that likely play a unique role in auditory processing.

Across brain regions, complex arrays of inhibitory interneurons sculpt the responses of excitatory neurons and dynamically regulate the local circuity. Therefore, diverse inhibitory cell types would be anticipated to exist in the VCN, one of the first stations of auditory signal processing. Thus, the VCN seemed quite different from this norm in that its only documented inhibitory cell, the D-stellate cell, is a projection neuron. Although previous staining and reconstruction studies have hinted at additional small, potentially local glycinergic neurons (*Lorente de Nó, 1981*; *Wenthold, 1987*; *Kolston et al., 1992*; *Doucet and Ryugo, 1997*; *Gleich and Vater, 1998*; *Doucet et al., 1999*), it has been a challenge to identify such cells based only on morphological and electrophysiological features. Due to their size, small cells are generally not targeted well for cell-filling procedures, especially *in vivo*. Electrophysiologically, the glycinergic L-stellate cells fire regularly, similar to excitatory T-stellate cells (*Oertel, 1983*; *Rhode et al., 1983b*; *Young et al., 1988*; *Oertel et al., 2011*), and thus could have occasionally been mistaken for T-stellate cells.

Morphological and electrophysiological features provide an initial platform for further classification of interneurons. When these approaches are applied to transgenic animals expressing different

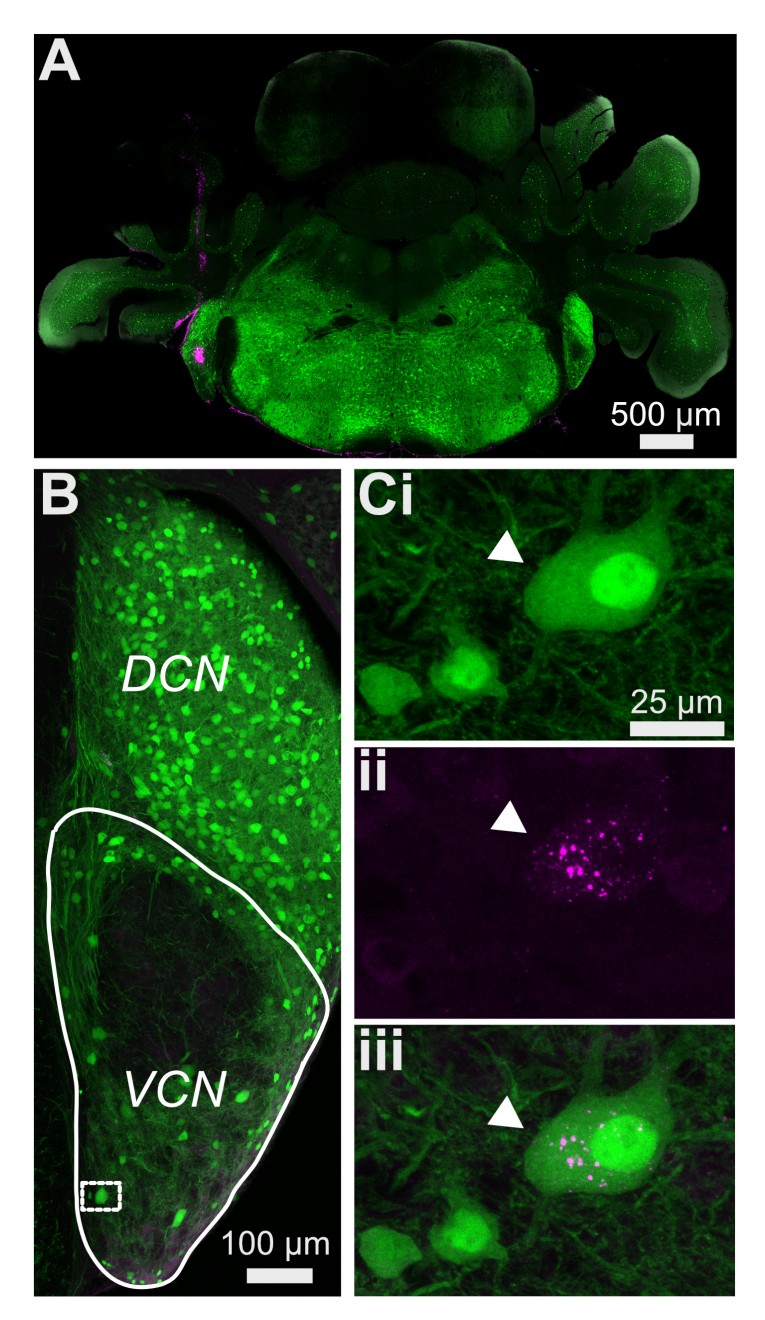

**Figure 9.** Small cells do not project contralaterally. (**A**) Retrogradely-transported fluorescent latex beads were injected into the contralateral CN of GlyT2-GFP mice. (**B**) VCN from the Ipsilateral CN was examined for neurons with retrobeads. (**C**) All the double-labeled neurons had large soma size (arrowhead) consistent with D-stellate cells. D-stellate cells (length of longest axis = 26.91 ± 3.92 μm, n = 47, n = 3 CNs). **Ci**, GFP; **Cii**, retrobeads; **Ciii**, merge.

molecular markers in inhibitory cell types, a clearer systematic classification system can emerge (*Wonders and Anderson, 2006*; *Cadwell et al., 2016*; *Fuzik et al., 2016*; *Tremblay et al., 2016*; *Pelkey et al., 2017*). In the SST-tdTomato::GlyT2-GFP line, large D-stellate cell types express both GFP and tdTomato, whereas smaller L-stellate cell types were only GFP-positive.

Although we have identified two major glycinergic interneuron populations, further studies may reveal additional diversity within the groups described in this paper. Our anatomical data show a

variety of somatic, dendritic, and axonal morphologies within the L-stellate cell types. Electrophysiologically, L-stellate cell types display heterogeneity in their membrane properties, likely a consequence of diverse ion channel expression. We also found that only a subset of L-stellate cells was sensitive to the cholinergic agonist, carbachol. Previous studies have reported that medial olivocochlear axons may make synaptic contacts with T-stellate cells and small unidentified cells in the VCN (*Benson and Brown, 1990*; *Benson et al., 1996*; *Fujino and Oertel, 2001*; *Oertel and Fujino, 2001*; *Mellott et al., 2011*). In addition to major cholinergic projections from the superior olivary complex, a substantial cholinergic input into CN is also reported from the pedunculopontine tegmental nucleus (PPT) and the latero-dorsal tegmental nucleus (LDT) (*Mellott et al., 2011*; *Schofield et al., 2011*). Cholinergic inputs to CN have been proposed to improve frequency selectivity, improve sound detection, or improve speech in noisy backgrounds (*Winslow and Sachs, 1987*; *Winslow and Sachs, 1988*; *Kawase et al., 1993*; *Fujino and Oertel, 2001*; *Mulders et al., 2002*). As some L-stellate cells were activated by bath application of carbachol, this suggests that the L-stellate cells might also receive inputs from cholinergic sources. L-cells may provide feedforward inhibition to control the excitability of T-stellate cells and bushy cells thereby regulating the signal detection in a noisy background. Single-cell transcriptomic analysis will likely allow further definition of subpopulations of glycinergic interneurons in the VCN (*Jiang et al., 2015*; *Cadwell et al., 2016*; *Fuzik et al., 2016*; *Shrestha et al., 2018*; *Sun et al., 2018*).

## Local inhibitory inputs in the VCN

In comparison to D-stellate cells, L-stellate cells had smaller soma sizes and exhibited tightly branched axons and dendrites. These restricted dendritic and axonal processes suggest that L-stellate cells have narrow frequency-band receptive fields. This morphology is very different from the D-stellate cell, which has been termed the 'wide-band inhibitor' (*Nelken and Young, 1994*; *Arnott et al., 2004*). The potential for narrow band excitation of the L-stellate cells could provide a means for inhibition to excitatory cells in nearby isofrequency lamina in VCN. Indeed, when we activated the smaller L-stellate cells with carbachol, we recorded IPSCs from both bushy and T-stellate cells. This provides evidence that L-stellate cells could in principle mediate side-band inhibition. Inhibition has been proposed to improve tuning properties and temporal precision, and stabilizes neuronal firing responses to acoustic inputs (*Gai and Carney, 2008*; *Keine and Rübsamen, 2015*). Indeed, as the D-stellate cells are numerically a minority of the glycinergic VCN cell population compared to the L-stellate cells, we propose that the majority of local inhibitory inputs onto excitatory neurons could originate from L-stellate cells.

D-stellate cells send their axons to the DCN and also project to contralateral VCN via the commissural pathway (*Schofield and Cant, 1996a*; *Doucet and Ryugo, 2006*). The commissural D-stellate cell connects ipsi- and contralateral CN. Previous tract tracing studies in rat reported a presence of small commissural glycinergic cells (*Doucet and Ryugo, 2006*), but we did not observe any L-stellate cells retrogradely labelled in our mouse study, consistent with a locally restricted axonal arbor. However, our cell-filling experiments only sampled a subset of cells, and it remains possible that some L-stellate cells also project outside the VCN, perhaps to the ipsilateral DCN or the granule cell region. Indeed, many L-stellate cells were located near the margins of VCN, near granule cell regions. These could be the small multipolar marginal cells described by *Doucet and Ryugo, 2006*. As inhibitory Golgi cells are located in and near the granule cell layer (*Mugnaini et al., 1980*; *Ferragamo et al., 1998b*; *Irie et al., 2006*; *Yaeger and Trussell, 2015*), it is possible that some L-stellate cells were Golgi cells.

In addition to local glycinergic sources, VCN receives projecting glycinergic inputs from the tuberculoventral cells of the DCN (*Wickesberg and Oertel, 1988*; *Wickesberg and Oertel, 1990*; *Wickesberg et al., 1991*; *Xie and Manis, 2013a*; *Campagnola and Manis, 2014*; *Muniak and Ryugo, 2014*). Tuberculoventral cells have narrow frequency-receptive fields and project to a tonotopically matched region in the VCN. Therefore, tuberculoventral cells have been proposed to provide narrow band inhibition, but at present, it is unclear whether the local glycinergic inputs from L-stellate cells and the long-range inputs from tuberculoventral cells play different roles in sculpting the responses of the excitatory VCN neurons. Conceivably, VCN principal cells could receive input from both subtypes of interneuron, and the differential activation and modulation of those interneurons could aid in fine tuning auditory processing under different conditions.

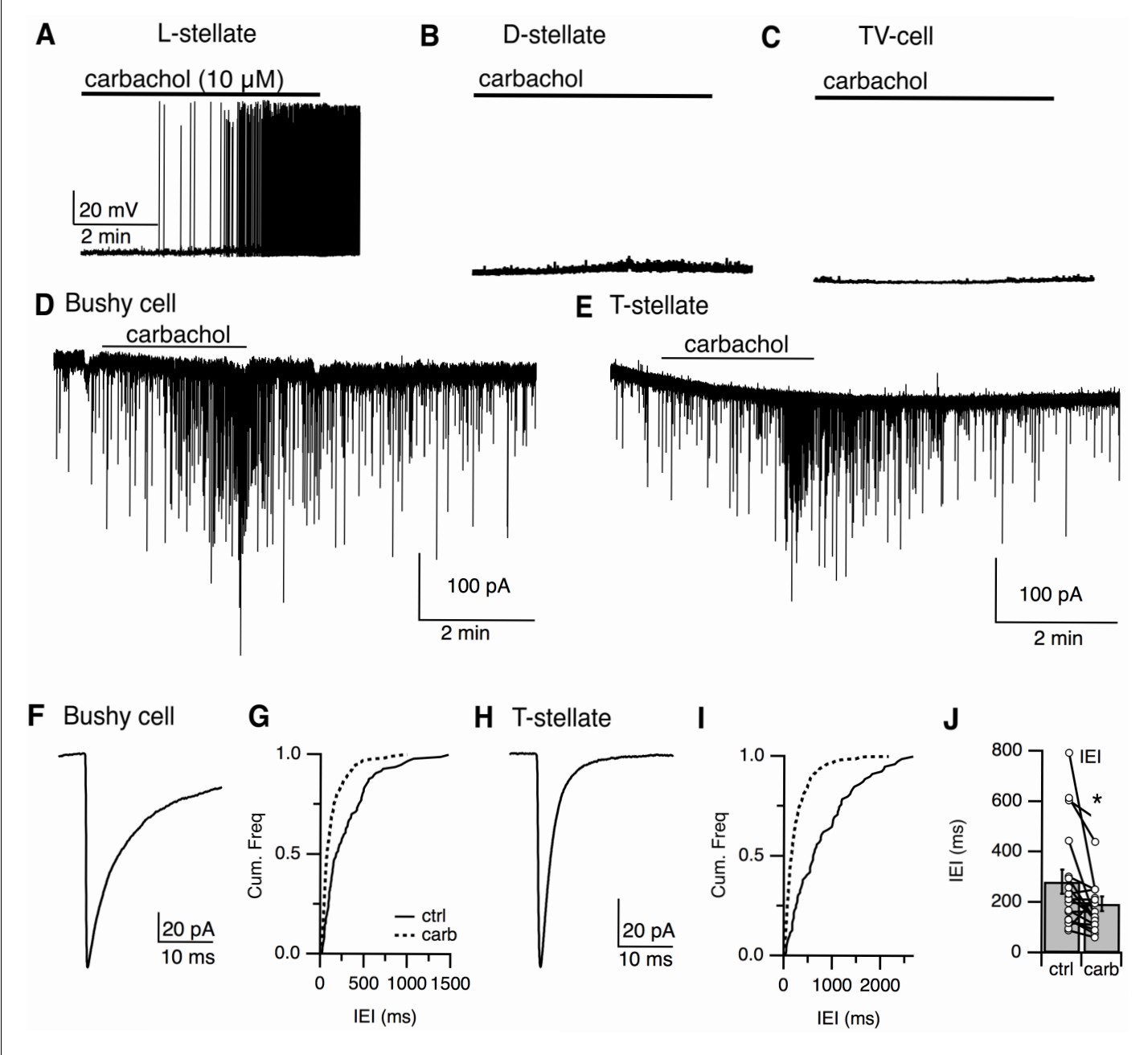

**Figure 10.** Local stellate cells (L-cells) provide local inhibition to primary cells in the VCN. (A–C) Representative responses to cholinergic agonist (carbachol, 10 µM) by small (L-cells) (A), D-stellate (B), tuberculoventral (TV) cells (C). Small cells were activated by carbachol. D-stellate and TV cells failed to respond to carbachol application. IPSCs in principal cells, bushy cells (D), and T-stellate cells (E) in the VCN were induced by carbachol application, confirming that small inhibitory neurons were local interneurons. (F, H) Time course of sIPSCs from bushy cells and T-stellate (bushy cell, τ = 7.22 ± 1.40 ms, n = 9, T-stellate, τ = 2.28 ± 0.07 ms, n = 9). (G, I) Cumulative histogram of inter-event internal under control and carbachol application. (J). Carbachol application resulted in a significant increase in sIPSCs frequency (inter-event interval, control = 280.44 ± 47.56 ms versus carbachol = 191.21 ± 28.99 ms, p<0.02, t-test).

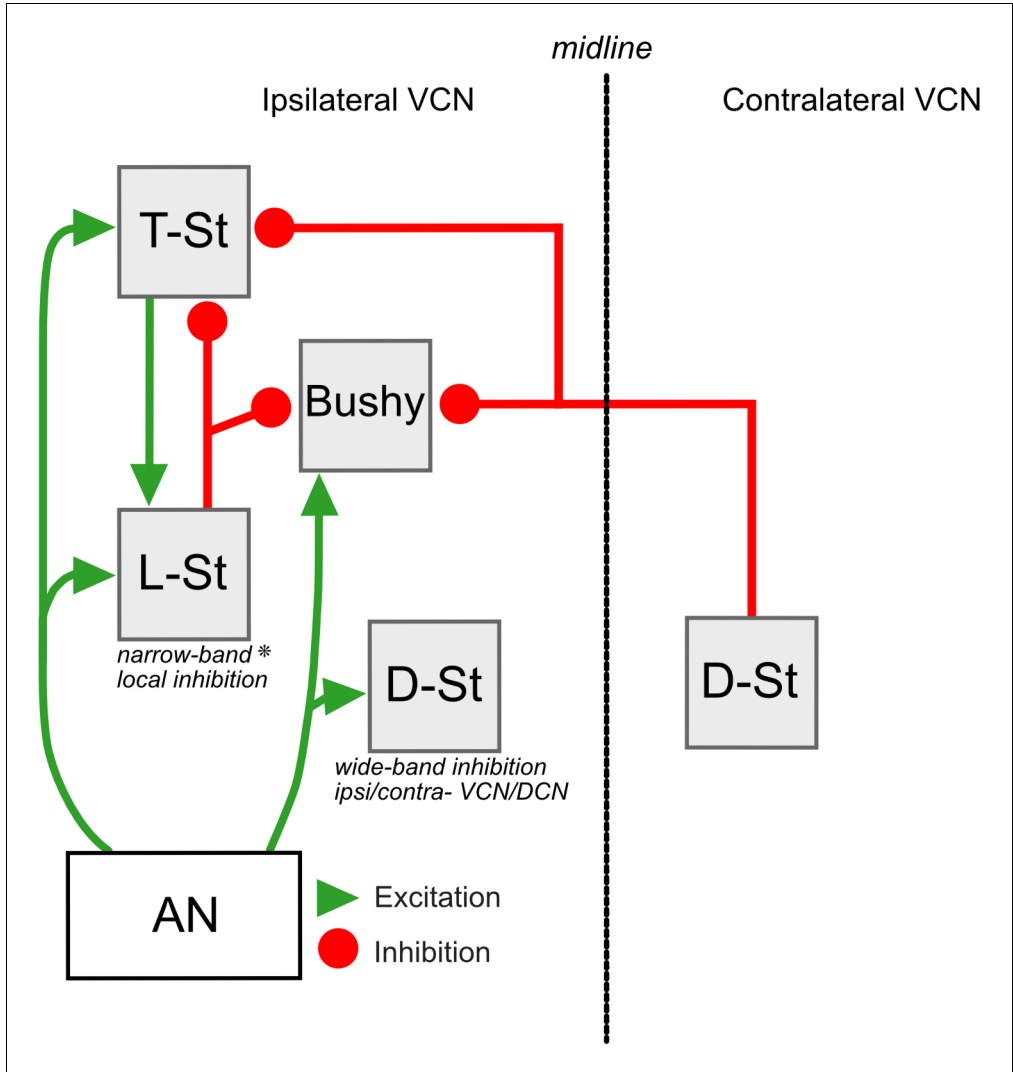

**Figure 11.** Summary of circuitry in the VCN. Neurons in the VCN including principal excitatory cells (T-stellate and bushy cells) and inhibitory cells (D-stellate and L-stellate cells) receive excitatory input from the auditory nerve (AN). L-stellate cells also receive polysynaptic inputs from T-stellate cells. L-stellate cells make local inhibitory synaptic contacts on principal cells of the VCN (* possibly function as narrow band local inhibitor). Principal cells also receive inhibition from D-stellate cells present in the contralateral VCN.

# Materials and methods

**Key resources table**

| Reagent type (species) or resource | Designation | Source or reference | Identifiers | Additional information |
|---|---|---|---|---|
| Strain, strain background (*M. musculus*) | C57BL/6J | Jackson Laboratory | RRID:JAX:000664 | |
| Strain, strain background (*M. musculus*) | GlyT2-EGFP, Tg(Slc6a5-EGFP)1Uze | MGI, *Zeilhofer et al., 2005* | RRID:MGI:3835459 | |
| Strain, strain background (*M. musculus*) | Sst-IRES-Cre, (Sst(tm2.1(cre)Zjh)) | MGI and Jackson Laboratory | RRID:MGI_4838416 RRID:JAX_013044 | |

*Continued on next page*

*Continued*

| Reagent type (species) or resource | Designation | Source or reference | Identifiers | Additional information |
|---|---|---|---|---|
| Strain, strain background (*M. musculus*) | Rosa 26, synonyms Ai9(RCL-tdT) | Jackson Laboratory PMID:22446880 | RRID:MGI_3809523 RRID:JAX_007909 | |
| Chemical compound, drug | Strychnine hydrochloride | Sigma | Cat# S8753 | |
| Chemical compound, drug | SR-95531 hydrobromide | Tocris | Cat# 1262 | |
| Chemical compound, drug | NBQX disodium salt | Tocris | Cat# 1044 | |
| Chemical compound, drug | (+)-MK-801 hydrogen maleate | Sigma | Cat# M107 | |
| Chemical compound, drug | Carbachol (carbamoylcholine chloride) | Sigma | Cat# C4382 | |
| Chemical compound, drug | Biocytin | ThermoFisher Scientific | Cat# B1592 | |
| Antibody | anti-GFP (Chicken polyclonal) | Aves Labs | Cat# GFP-1020 RRID:AB_10000240 | IHC (1:2000) |
| Antibody | anti-DsRed (Rabbit polyclonal) | Clontech | Cat# 632496 RRID:AB_10013483 | IHC (1:1000) |
| Antibody | anti-Glycine (Rabbit polyclonal) | Millipore Sigma | Cat# Ab139 RRID:AB_90582 | IHC (1:500) |
| Antibody | anti-rabbit Cy3 (Donkey polyclonal) | Jackson Immuno Research Labs | Cat# 711-165-152 RRID:AB_2307443 | IHC (1:500) |
| Antibody | anti-chicken Alexa Fluor 488 (Donkey polyclonal) | Jackson Immuno Research Labs | Cat# 703-545-155 RRID:AB_2340375 | IHC (1:500) |
| Antibody | anti-Rabbit Alexa Fluor 594 (Donkey polyclonal) | Jackson Immuno Research Labs | Cat# 711-585-152 RRID:AB_2340621 | IHC (1:400) |
| Chemical compound, drug | Tetrakis (N,N,N',N'-tetrakis (2-hydroxypropyl) ethylenediamine) | Sigma | Cat# 122262 | |
| Chemical compound, drug | Red Retrobeads | LumaFluor. Inc | Red Retrobeads | |
| Software, algorithm | pClamp 10 | Molecular Devices | RRID:SCR_011323 | |
| Software, algorithm | Igor Pro 8 | WaveMetrics | RRID:SCR_000325 | |
| Software, algorithm | NeuroMatic | *Rothman and Silver, 2018*; DOI: 10.3389 | RRID:SCR_004186 | |
| Software, algorithm | Axograph | Axograph | RRID:SCR_014284 | |
| Software, algorithm | Prism 5 | GraphPad | RRID:SCR_002798 | |
| Software, algorithm | Excel | Microsoft | RRID:SCR_016137 | |
| Software, algorithm | Imaris | Bitplane | RRID:SCR_007370 | |
| Software, algorithm | Zen Black | Zeiss | RRID:SCR_013672 | |
| Software, algorithm | ImageJ | National Institute of Health, ImageJ.net | RRID:SCR_003070 | |
| Software, algorithm | FIJI | Fiji.sc | RRID:SCR_002285 | |
| Software, algorithm | Affinity Designer | Serif | RRID:SCR_016952 | |
| Software, algorithm | Adobe Illustrator | Adobe | RRID:SCR_010279 | |

## Animals

All procedures were approved by the Oregon Health and Science University's Institutional Animal Care and Use Committee. GlyT2-GFP mice (*Zeilhofer et al., 2005*) and somatostatin (SST)-Cre tdTomato::GlyT-GFP of either sex, postnatal days(P) 17–30 were used in this study. GlyT2-GFP mice were backcrossed into the C57BL/6J and maintained as heterozygous (*Roberts et al., 2008*). SST-tdTomato::GlyT2-GFP mice were generated as follows: Homozygous SST-IRES-Cre knock-in mice (Jackson Laboratory) were crossed with homozygous Ai9 (RCL-tdTomato) reporter mice (Jackson Laboratory), enabling SST-IRES-Cre-dependent expression of tdTomato. Offspring from the F1 generation (SST-IRES-Cre+/-, Ai9(RCL-tdTomato)+/-) were crossed with homozygous GlyT2-GFP mice, whose offspring will be referred to as SST-tdTomato::GlyT2-GFP. At 5 days postnatal, a fluorescent stereoscope (Leica Microsystems) was used to identify transgenic mice positive for either tdTomato or GFP.

## Brain-slice preparation

Animals were anesthetized with isoflurane and decapitated. The brain was quickly removed and placed into ice-cold sucrose cutting solution. Sucrose solution contained (in mM) 76 NaCl, 26 NaHCO$_3$, 75 sucrose, 1.25 NaH$_2$PO$_4$, 2.5 KCl, 25 glucose, 7 MgCl$_2$, and 0.5 CaCl$_2$, bubbled with 95% O$_2$: 5% CO$_2$ (pH 7.8, 305 mOsm). Parasagittal slices of CN were cut at a slight angle from sagittal, to best preserve a straight projection of the AN (*Ngodup et al., 2015*). The slices of CN were cut at 220–300 μm in ice-cold sucrose solution on a vibratome (Leica VT1200S or Campden 7000smz-2). Slices were transferred into standard artificial cerebrospinal fluid (ACSF) containing (in mM) 125 NaCl, 26 NaHCO$_3$, 1.25 NaH$_2$PO$_4$, 2.5 KCl, 20 glucose, 1 MgCl$_2$, 1.5 CaCl$_2$, 2 Na-pyruvate, and 0.4 Na L-ascorbate, bubbled with 95% O$_2$:5% CO$_2$ (pH 7.4, 300–310 mOsm). Slices recovered at 34°C for 40 min and were maintained at room temperature until recording.

## Electrophysiology

Slices were transferred to a recording chamber and perfused with standard ACSF (~34°C). Cells were viewed using an Olympus BX51WI microscope with a 60× objective, equipped with an infrared Dodt contrast, CCD camera (Retiga 2000R, QImaging), and fluorescence optics. In slices from GlyT2-GFP, glycinergic cells in the VCN were identified by their GFP expression. Recording pipettes were pulled from 1.5 mm OD, 0.84 mm ID borosilicate glass (WPI-1B150-F) to a resistance of 2–4 MΩ using a horizontal puller (Sutter Instrument P97). The internal recording solution contained in (mM) 113 K gluconate, 2.75 MgCl$_2$, 1.75 MgSO$_4$, 0.1 EGTA, 14 Tris$_2$-phosphocreatine, 4 Na$_2$-ATP, 0.3 Tris-GTP, 9 HEPES with pH adjusted to 7.25 with KOH, mOsm adjusted to 290 with sucrose (E$_{Cl}$−84 mV). For a few voltage-clamp experiments, we used an internal solution containing (in mM) 115 CsCl, 1 MgCl$_2$, 4 Mg-ATP, 0.4 Tris-GTP, 5 EGTA, 14 Tris$_2$-phosphocreatine, 4 Na$_2$-ATP, 10 HEPES, 3 QX-314 (pH 7.2, 290 mOsm). Whole cell patch-clamp recordings were made using a Multiclamp 700B amplifier and pCLAMP 10 software (Molecular Devices). Signals were digitized at 20–40 kHz and filtered at 10 kHz by Digidata 1440A (Molecular Devices). In voltage clamp, cells were held at −70 mV, with access resistance 5–20 MΩ compensated to 50–60%. In the current clamp, the resting membrane potential was maintained at −60 to −70 mV with bias current. To isolate excitatory postsynaptic currents, inhibitory synaptic blockers SR-95531 (10 μM) and strychnine (0.5 μM) were added to the bath solution whereas to isolate inhibitory currents, excitatory synaptic blockers NBQX (10 μM) and MK-801 (5 μM) were added to the bath solution. The AN root was stimulated with brief voltage pulses (100 μs) using a stimulus isolation unit (Iso-Flex, A.M.P.I) via a bipolar microelectrode placed in the nerve root.

## Electrophysiological analysis

Spike waveforms were analyzed using Igor Pro 8. Threshold was defined as the voltage at 5% of the peak of the spike's first derivative, measured after subtracting the baseline dV/dt generated by the neuron's electronic charging. This method was found empirically to identify the visually identified spike 'onset' much more reliably than the oft-used peak of the 3rd derivative (*Kuo et al., 2012*). The latter method, when applied to diverse neurons in our dataset, often failed when the spike had a prominent initial segment 'hump' on the rising phase, or when the 3rd derivative was noisy. Spike overshoot was calculated as spike peak voltage relative to zero. Absolute amplitude of the spike

was calculated from threshold to peak. Undershoot was the negative-most excursion of voltage, relative to threshold, and AHP latency was the time delay from spike peak to undershoot peak. Spike AHP was an arbitrary definition that measured the difference between voltage at the peak of the undershoot to the voltage one ms later. This time point captured well differences among neurons, since the D-stellate cells, and not the smaller cells, had extremely fast decaying AHPs. Membrane input resistance ($R_{in}$) was measured by injecting a small hyperpolarizing current in voltage clamp mode, $R_{in}$ was calculated offline using Ohm's law. Average spike rate was calculated in response to depolarizing 500 pA current injections.

## Morphology

Morphological studies of individual cells were made by including 0.2–0.4% biocytin (B1592, Molecular Probes) in the recording pipette solution. After loading each cell for 20 min, the recording electrode was slowly retracted, and the slice fixed overnight in 4% (w/v) buffered paraformaldehyde. After fixation, the slices were rinsed in PBS and stored for up to a week at 4°C in PBS until processing for biocytin labeling. To visualize biocytin labeling, the slices were permeabilized in 0.2% Triton X-100 solution (in PBS) for 2 hr at room temperature. Slices were incubated in 0.3% $H_2O_2$ for 30 min to quench endogenous peroxidase, rinsed with PBS, incubated in ABC reagent (Vector Laboratories) for 2 hr, rinsed with PBS, and then incubated for 3–4 min in diaminobenzidine (DAB) solution containing 0.05 M Tris buffer, 10 mg/mL nickel ammonium sulfate, 50 mM imidazole, 1 mg (1 mg/100 μL) DAB, and 0.3% $H_2O_2$. Slices were then rinsed, mounted on glass slides, dehydrated in an ascending series of alcohols, delipidized in xylene, and cover slipped with Permount. DAB-stained cells were visualized with a Zeiss Axio Imager M2 using a 40× oil immersion objective and reconstructed using Neurolucida (MBF Bioscience). Analyses of dendritic and axonal processes were made in Neurolucida explorer.

## Immunohistochemistry

For glycine labeling, mice were deeply anesthetized with isoflurane and perfused transcardially with 0.9% warm saline followed by a fixative containing 2% glutaraldehyde and 1% paraformaldehyde buffered in PBS. Brains were removed and post-fixed for 30–60 min. Sagittal sections were cut at 50 μm thickness on a vibratome (Leica VT1000S). Slices were rinsed in PBS and then incubated in fresh 1% NaBH₄ for 30 min to reduce autofluorescence from glutaraldehyde fixation. Slices were rinsed extensively in PBS.

For colabeling studies of SST-tdTomato and GlyT2-GFP-positive neurons, SST-tdTomato::GlyT2-GFP mice were deeply anesthetized with isoflurane and perfused transcardially with 0.9% warm saline followed by 4% paraformaldehyde buffered in PBS. Brains were removed and post-fixed overnight at 4°C. Sections were cut at 50 μm thickness on a vibratome (Leica VT1000S).

All sections were then blocked in 2% bovine serum albumin (BSA), 2% fish gelatin, and permeabilized in 0.2% Triton X-100 in PBS for 2 hr at room temperature on a 2-D shaker table. Next, the sections were incubated in primary antibody solution containing: primary antibody, 2% BSA, 2% fish gelatin PBS with 0.2% Triton X-100 for 24–48 hr at 4°C. The sections were then washed three times, 10-mins each, in PBS, and incubated in secondary antibody solution for 2 hr at room temperature or 24–48 hr at 4°C on a shaker table. The following primary antibodies were used: 1:500 anti-glycine (Ab139, Millipore Sigma), 1:2000 chicken anti-GFP (GFP-1020, Aves Labs), and 1:1000 rabbit anti-DsRed (632496, Clontech). The following secondary antibodies were used: 1:500 Cy3-conjugated donkey anti-rabbit antibody (Jackson Immuno Research Labs), 1:500 donkey anti-chicken conjugated to Alexa Fluor 488 (703-545-155, Jackson Immuno Research Labs), and 1:400 donkey anti-rabbit conjugated to Alexa Fluor 594 (711-585-152, Jackson Immuno Research Labs). Sections were rinsed in PBS three times, 10 min for each rinse, and in a few cases, sections were subsequently incubated in 4% paraformaldehyde in PBS for 1 hr. Sections were then mounted on microscope slides and cover-slipped with Fluoromount-G (Southern Biotech). Images were acquired using a Zeiss LSM 780 confocal microscope.

## Thick tissue clearing

Mice were deeply anesthetized with isoflurane, then perfused transcardially with 0.9% warm saline followed by 4% paraformaldehyde buffered in PBS. Brains were removed and post-fixed overnight

at 4°C. The entire CN (450–500 µm) was cut from the brainstem using a vibratome (Leica VT1000S). Because the CN sits at the lateral edge of the brainstem, only a single cut was required to prepare the specimen. Samples were washed in 0.1 M PBS before clearing for 72 hr at room temperature on a shaker table with an accelerated 'clear unobstructed brain imaging cocktail' (CUBIC)-mount solution (*Lee et al., 2016*) containing sucrose (50%, w/v), urea (25%, w/v), and N,N,N′,N′-tetrakis (2-hydroxypropyl) ethylenediamine (25%, w/v) dissolved in 30 mL of dH$_2$O. Once each sample was cleared and almost transparent, it was then mounted on a glass slide with 0.5 mm deep silicone spacers (EMS) in CUBIC-mount. The images were acquired within 24 hr to reduce volume changes in CUBIC-mount (10–20% increase in volume after incubation). The samples were imaged using a Zeiss LSM880 with Airyscan microscope with a 25× objective. Tiles and z-stack of whole CN were imaged at 5 µm step size with the exception of one sample where images were obtained at 2 µm step size. Images were post-processed and individual tiles were stitched together using Zeiss Zen Black software. Analysis of cell count and soma volume were quantified using Imaris software (Bitplane 12.1). For optimal processing of large file sizes, files were separated into 3–4 smaller files each containing (30 GB)~150 µm of CN for faster processing. The image stacks were further processed for background correction and normalization in Imaris. VCN was selected for quantification by drawing regions of interest around the VCN followed by applying a mask outside the region of interest. The dorsal border of the VCN was defined by the granule cell lamina, and the medial border by the absence of glycinergic somata (*Muniak et al., 2013*). For cell count quantification, we used the built-in 'spot detection' algorithm in Imaris in which the program places a 'spot' on the soma of each cell. The spots are used for counting of GFP positive cells in the VCN. Automatically detected spots were verified by the eye and corrected manually. To test the validity of the automated spot function, certain regions of VCN were manually counted and compared against the spot function. The spot function detected more than 95% of all cells present in that area. For volume measurements, we used the in-built 'surface' rendering function in Imaris. The program rendered 3D surfaces on the soma of GFP positive cells. Rendered surfaces were used to extract the volume statistics. Surfaces were also verified and corrected manually.

## Colocalization assay

For colocalization analysis of tdTomato with GFP in the VCN, images were obtained from CUBIC-cleared VCN of SST-tdTomato::GlyT2-GFP transgenic mice. Images were acquired at 5 µm steps with Zeiss LSM880 with Airyscan as described above. Images were post-processed and stitched using Zeiss Zen Black software. Images were analyzed using the 'ImarisColoc' function in Imaris. 150 µm thickness of VCN was processed separately. ImarisColoc function allows to process the overlap between the two-color channels in an image. Minimum threshold was selected for the two-color channels. A new channel was generated that only contained colocalized voxels. Next, we used the surface rendering program, which turns voxels into solid objects, which were used to measure the volumes on double-labeled cells.

## Retrobead injections

GlyT2-GFP mice were anesthetized with isoflurane and placed in a stereotaxic frame (David Kopf). Animal temperature was maintained near 37°C with a heating pad (T/pump Gaymar). The scalp was retracted, a portion of the skull above the left cerebellum was opened. VCN was located by stereotactic coordinates starting from the surface junction point of the inferior colliculus, cerebellar lobule IV-V, and simple lobule (0.7 mm lateral, 0.95 mm rostral, 4 mm depth). Glass capillaries (Wire Trol II, Drummond Scientific) were pulled on a horizontal puller (P-97, Sutter) and then beveled using a diamond lapping disc (0.5 µm grit, 3M DLF4XN_56611X) to an inside diameter of 20–30 µm (*Balmer and Trussell, 2019*). Glass capillaries were advanced into the VCN with a microdrive (IVM-500, Scientifica) at a rate of 10 µm/s. 50 –100 nL of red retrobeads (LumaFluor Inc) was injected using a single axis manipulator (MO-10, Narishige) and pipette vice (Ronal). 5 min waiting periods were allowed before and after injections. The skin was sutured and mice were allowed to recover for 5–6 days. After recovery, mice were deeply anesthetized with isoflurane and then perfused transcardially with 0.9% saline followed by 4% paraformaldehyde buffered in PBS. Brains were removed and post-fixed overnight at 4°C. Coronal sections were cut at 50 µm thickness on a vibratome (Leica

VT1000S). Confocal images of the sections were acquired using a Zeiss LSM 780 confocal microscope. Images were analyzed using ImageJ.

## Pharmacology

All drugs in the slice experiments were bath applied. Receptor antagonists used in this study included: NBQX (AMPA receptors: Sigma), MK-801 (NMDA receptors; Sigma), SR-95531 (GABA$_A$R; Tocris), strychnine (glycine receptors; Sigma). Acetylcholine receptors were activated using the non-selective cholinergic agonist, carbamoylcholine chloride (carbachol) (Sigma).

## Experimental design and statistical analyses

Electrophysiological data were analyzed using pClamp 10.4 software (Molecular Devices), Axograph, or IGOR Pro v6.3 or v8 (WaveMetrics) and NeuroMatic (*Rothman and Silver, 2018*). Figures were made using IGOR Pro, Affinity Designer, and Adobe Illustrator. Statistics were performed in IGOR Pro, Axograph, Python, Microsoft Excel, or Prism (GraphPad). For statistical analysis, groups were compared with paired or unpaired t-test. Cluster analysis was performed using sklearn.cluster. KMeans in Python and figures were made using matplotlib.pyplot. Error bars are represented as mean ± SEM unless otherwise stated.

## Acknowledgements

We thank Ben Suter, Taro Kiritani, and Carl Peterson for permission and assistance with the CUBIC protocol, Stefanie Kaech Petrie, Aurelie Snyder, Crystal Chaw, and Brian Jenkins from the Advanced Light Microscopy Core, The Jungers Center for assistance with microscopy, T Balmer, T Garett, H Hong, L Moore, J Tang, D Zeppenfeld for helpful comments on the manuscript. We also want to thank M J Murdock Charitable Trust for endowing funds for microscopes at the OHSU Microscopy Core. These experiments were supported by Hearing Health Foundation's Emerging Research Grant to TN, National Institute of Health (NIH) Grants R01 NS028901 and DC004450 to LOT, NINDS P30NS061800 to Imaging Center, OHSU. Gabriel E Romero is a Howard Hughes Medical Institute Gilliam Fellow. We wish to dedicate this study to the late Dr. Donata Oertel, who loved the stellate cells of the VCN.

## Additional information

### Funding

| Funder | Grant reference number | Author |
|---|---|---|
| National Institutes of Health | NS028901 | Laurence O Trussell |
| National Institutes of Health | DC004450 | Laurence O Trussell |
| National Institutes of Health | P30NS061800 | Laurence O Trussell |
| Hearing Health Foundation | Emerging Research Grant | Tenzin Ngodup |
| Howard Hughes Medical Institute | Gilliam Fellowship | Gabriel E Romero |
| National Institutes of Health | NS116798 | Laurence O Trussell |

The funders had no role in study design, data collection and interpretation, or the decision to submit the work for publication.

### Author contributions

Tenzin Ngodup, Conceptualization, Data curation, Software, Formal analysis, Funding acquisition, Validation, Investigation, Methodology, Writing - original draft, Writing - review and editing; Gabriel E Romero, Resources, Investigation, Methodology, Writing - review and editing; Laurence O Trussell, Conceptualization, Formal analysis, Supervision, Funding acquisition, Writing - original draft, Project administration, Writing - review and editing

## Author ORCIDs

Laurence O Trussell (iD) https://orcid.org/0000-0003-1171-2356

## Ethics

Animal experimentation: This study was performed in strict accordance with the recommendations in the Guide for the Care and Use of Laboratory Animals of the National Institutes of Health. All experimental procedures were approved by the Oregon Health and Science University's Institutional Animal Care and Use Committee, under protocol IP00000952.

## Decision letter and Author response

Decision letter https://doi.org/10.7554/eLife.54350.sa1
Author response https://doi.org/10.7554/eLife.54350.sa2

## Additional files

### Supplementary files

• Transparent reporting form

### Data availability

Datasets have been uploaded to Dryad, with the DOI doi:10.5061/dryad.69p8cz8xp These are referred to in the appropriate figure legends.

The following dataset was generated:

| Author(s) | Year | Dataset title | Dataset URL | Database and Identifier |
|---|---|---|---|---|
| Trussell L | 2020 | Discovery of a novel inhibitory neuron class, the L-Stellate cells of the cochlear nucleus | http://dx.doi.org/10.5061/dryad.69p8cz8xp | Dryad Digital Repository, 10.5061/dryad.69p8cz8xp |

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
