## [Decision Letter]

**Acceptance summary:**

This report describes the discovery and key properties of an additional class of glycinergic interneuron in the mouse ventral cochlear nucleus, the site at which auditory information is first processed in the brain. The authors demonstrate the existence of these cells using a GlyT2-GFP reporter mouse and perform a morphometric analysis to define their size and dendritic arborizations, noting that the cells are smaller than the previously described D-stellate cells. The cells (termed "L-stellate" cells) appear to represent a distinct class, as they are not labeled in SST-Cre mice, they do not appear to send projections to the contralateral cochlear nucleus, and they exhibit distinct morphological and electrophysiological properties. These cells appear to receive feedforward excitation from auditory nerve fibers, and using a pharmacological approach, the authors provide evidence that these neurons provide inhibitory input to both bushy calls and T-stellate cells. L-stellate cells represent the most numerous inhibitory cells in the ventral cochlear nucleus, suggesting that they play a critical role in processing sound information.

**Decision letter after peer review:**

Thank you for submitting your article "Discovery of a novel inhibitory neuron class, the L-Stellate cells of the cochlear nucleus" for consideration by *eLife*. Your article has been reviewed by three peer reviewers, including Dwight E Bergles as the Reviewing Editor and Reviewer #1, and the evaluation has been overseen by Barbara Shinn-Cunningham as the Senior Editor.

The reviewers have discussed the reviews with one another and the Reviewing Editor has drafted this decision to help you prepare a revised submission.

[…]

The manuscript is very clearly written, and the experiments performed with a high degree of rigor, as expected from this outstanding group. The results are novel and important for the auditory field. However, all reviewers raised substantive major concerns, which preclude acceptance of the manuscript in its present form. There was a consensus that, while interesting for auditory neuroscientists, the studies do not go far enough to reveal how these cells influence information flow through the CN and thus many not be of broad enough interest to warrant publication in *eLife*. However, it the studies could be expanded to better define the role of these interneurons in the inhibitory control of auditory coding in the cochlear nucleus, the impact and interest would be substantially higher and thus appropriate for the readership of *eLife*. The reviewers recommend addressing the following four major comments.

1) Relationship to previous findings: There is preceding data in the literature from several other species that revealed the presence of a set of small inhibitory interneurons in the VCN (glycinergic, and possibly also GABAergic) that are not part of the commissural pathway (Gleich and Vater, 1998; Kolston et al., 1992; Doucet and Ryugo, 1997; Doucet et al., 1999, etc). These observations are not often emphasized in the literature, probably because they are scattered and are not included in any of the standard conceptual frameworks about the CN circuit. However, here they should be acknowledged more clearly in the manuscript, both in the Introduction and Discussion. The L-stellate cells are not really "novel" (as stated in the Abstract and title), although the ability to target them for study, and revealing part of their position and function as inhibitory interneurons in the circuit, is a significant and important advance. What is most remarkable is the number of cells in this class, and that they have been "missed", or at least not recognized, in most previous physiological studies. Thus, the title of the study should be adjusted to reflect this earlier work.

2) Role of L-stellate cells in the DCN: The main drawback of the study is that the functional impact of local inhibition vs. previously known sources of tonotopic inhibition from the DCN remains unclear, apart from differential sensitivity to cholinergic modulation. It is not apparent from the data presented how strong L-stellate connections are on their VCN targets, or their dynamics in response to ANF stimulation. The absence of paired recordings is a major weakness – does this reveal something unusual about their connections? Perhaps this could be addressed by using another approach – cell attached screening or using slices with a different orientation, etc. Without these data, or more detailed histological analysis (similar to what has been done at the EM level for interneurons in the hippocampus) it is difficult to place the cells in a functional context.

One of the main questions this study raises is what is the functional distinction of L-stellate vs. DCN (vertical cell) mediated tonotopic inhibition to VCN targets. Is L-stellate cell input functionally redundant? Can the authors preferentially stimulate this inhibition from the nerve without the DCN? Can the strength and dynamics of inhibition during repetitive stimulation from nerve stimulation be observed in the coronal plane in which the L-stellate cells are the only intact inhibitory cells in the slice? Some comparison with the T-stellate cells is warranted (perhaps in conjunction with Figure 5, in terms of dendritic and axonal morphology). This does not need to be an exhaustive comparison, just enough to help clarify if there are any pronounced differences. In this regard, the L-stellate dendritic fields look a lot like previous reconstructions of T-stellate cells, so how different are those 2 classes? How likely is it that previous studies claiming to record from T-stellate cells have actually mixed the populations?

3) Morphological characteristics of L-stellate cells: The axons of L-stellate cells are not clearly delineated in the experimental material. The drawings in Figure 5 show "axonal and dendritic arbors", but do not indicate which of the traced processes belong to which part of the cell. Although the data support the conclusion that these cells do not project contralaterally in any major way, it is unclear whether they might have other projections, such as to the DCN (it does not appear so from the reconstructions, but that is not a foregone conclusion as it is unclear how visible the axon was or whether it could be followed to its terminations). The "localness" of these cell's targets should be better qualified. In Figure 5, the heterogeneity of dendritic morphology among L-stellate cells is striking. Does dendritic morphology reflect electrophysiological differences?

4) Physiological properties of L-stellate cells: The demonstration of chopping response to direct current injection in Figure 7 might be expected for any regularly firing neuron. It would be more informative if the authors could demonstrate chopping in response to ANF stimulation. The diversity in input resistance and spike parameters across cells (e.g., in Figure 6) might have interesting functional consequences in these experiments and might be distinct from vertical cell responses. As the authors point out, it seems likely that there are at least two classes of L-stellate cells, but their different physiological characteristics have not been related to differences in morphology (Figure 5 and 6), so it is difficult to place them in a relevant context. It would be helpful if the authors could provide more information on this issue from their existing data set. Does carbachol sensitivity correlate with differences in AP shape parameters or topographic location within the VCN? In other words, does carbachol sensitivity reflect differences in cell type or simply the location of the cell with respect to cholinergic input fields?

[Editors' note: further revisions were suggested prior to acceptance, as described below.]

Thank you for resubmitting your work entitled "Identification of an Inhibitory Neuron Subtype, the L-Stellate Cell of the Cochlear Nucleus" for further consideration by *eLife*. Your revised article has been evaluated by Barbara Shinn-Cunningham (Senior Editor) and a Reviewing Editor.

The reviewers agree that the manuscript has been substantially improved, despite the current limitations due to COVID-19, and that the work is appropriate for publication in *eLife*. However, several remaining issues need to be addressed before final acceptance, as outlined below:

Additional quantitative information should be included, if possible, about the dendritic morphology of L- vs. T-stellate cells. In Figure 5—figure supplement 1: the analyses of L- vs. T-stellate cell dendritic morphology appears anecdotal, with just one L-stellate and 3 T-stellate cells reconstructed. This should be better documented, as this is a substantial part of the comparison between excitatory and inhibitory stellate cells in the VCN. It seems feasible even in a reduced research state to flesh out this data set enough to provide a statistical comparison.

Please provide a small orientation indicator for all figures with anatomical images. If all are in the same orientation (they appear to be), then this should be clearly stated somewhere in the manuscript. While I can figure out what the orientation is from clues in the sections, other readers could have difficulty.

Figure 5: add statistical information for the added panel F legend, to be consistent with the rest of the legend.

Figure 6—figure supplement 1: There is no reference to this information in the main text. Please add any statistical information to the figure legend.

Introduction paragraph two: Lorente de No could not know neurons were "small inhibitory neurons", but he did describe many small cells in the VCN. Please reword (you could just remove "inhibitory" from the sentence).

Introduction paragraph three: "receptive field" is still misleading here. They have a narrower dendritic field, from which a narrower receptive field would be inferred. Perhaps reword.

Introduction paragraph three : "excitability of excitatory neurons " suggest -> "excitability of projection neurons "

Results paragraph three: the link is incorrect: it is "http://mouse.brain-map.org/experiments/show/69874024".

Subsection “Synaptic connectivity”: Just to reinforce this observation, delayed firing after trains of AN stimulation were also reported in a subset of "planar multipolar" cells by Xie and Manis, 2017. However, it is unclear whether the cells in that study correspond to the L-stellate cells in this study as the cell identification was not as robust.

Paragraph two in subsection “Synaptic connectivity”: number agreement: "cells have" (not "cells has")

"For a few voltage-clamp experiments, we used an internal solution containing (in mM) 5 CsCl, 1 MgCl2, 4 Mg-ATP, 0.4 Tris-GTP, 5 EGTA, 14 Tris2-phosphocreatine, 4 Na2-ATP, 10 HEPES, 3 QX-314 (pH 7.2, 290 mOsm)." It is not clear to me how this electrode solution has an osmolarity of 290 mOsm. I still think there is something missing in this description. Please check.

---

## [Author Response]

The manuscript is very clearly written, and the experiments performed with a high degree of rigor, as expected from this outstanding group. The results are novel and important for the auditory field. However, all reviewers raised substantive major concerns, which preclude acceptance of the manuscript in its present form. There was a consensus that, while interesting for auditory neuroscientists, the studies do not go far enough to reveal how these cells influence information flow through the CN and thus many not be of broad enough interest to warrant publication in eLife. However, it the studies could be expanded to better define the role of these interneurons in the inhibitory control of auditory coding in the cochlear nucleus, the impact and interest would be substantially higher and thus appropriate for the readership of eLife. The reviewers recommend addressing the following four major comments.1) Relationship to previous findings: There is preceding data in the literature from several other species that revealed the presence of a set of small inhibitory interneurons in the VCN (glycinergic, and possibly also GABAergic) that are not part of the commissural pathway (Gleich and Vater, 1998; Kolston et al., 1992; Doucet and Ryugo, 1997; Doucet et al., 1999, etc). These observations are not often emphasized in the literature, probably because they are scattered and are not included in any of the standard conceptual frameworks about the CN circuit. However, here they should be acknowledged more clearly in the manuscript, both in the Introduction and Discussion. The L-stellate cells are not really "novel" (as stated in the Abstract and title), although the ability to target them for study, and revealing part of their position and function as inhibitory interneurons in the circuit, is a significant and important advance. What is most remarkable is the number of cells in this class, and that they have been "missed", or at least not recognized, in most previous physiological studies. Thus, the title of the study should be adjusted to reflect this earlier work.

We appreciate that the reviewer sees that this work has made a significant and important advance. We have revised our manuscript to explicitly acknowledge previous studies reporting the presence of small inhibitory neurons that are distinct from D-stellate cells, Introduction paragraph three. The word “novel” in the title and manuscript was used to describe the majority of unknown small inhibitory neurons that were not reported earlier. We have nevertheless changed the title to “Identification of an inhibitory neuron subtype, the L-stellate cell of the cochlear nucleus.” However, as the reviewers likely know, these cells are indeed novel to the extent that previous proposals about the key elements of circuitry and computation in the CN have ignored them, despite occasional reports their appearance.

2) Role of L-stellate cells in the DCN: The main drawback of the study is that the functional impact of local inhibition vs. previously known sources of tonotopic inhibition from the DCN remains unclear, apart from differential sensitivity to cholinergic modulation. It is not apparent from the data presented how strong L-stellate connections are on their VCN targets, or their dynamics in response to ANF stimulation. The absence of paired recordings is a major weakness – does this reveal something unusual about their connections? Perhaps this could be addressed by using another approach – cell attached screening or using slices with a different orientation, etc. Without these data, or more detailed histological analysis (similar to what has been done at the EM level for interneurons in the hippocampus) it is difficult to place the cells in a functional context.One of the main questions this study raises is what is the functional distinction of L-stellate vs. DCN (vertical cell) mediated tonotopic inhibition to VCN targets. Is L-stellate cell input functionally redundant? Can the authors preferentially stimulate this inhibition from the nerve without the DCN? Can the strength and dynamics of inhibition during repetitive stimulation from nerve stimulation be observed in the coronal plane in which the L-stellate cells are the only intact inhibitory cells in the slice? Some comparison with the T-stellate cells is warranted (perhaps in conjunction with Figure 5, in terms of dendritic and axonal morphology). This does not need to be an exhaustive comparison, just enough to help clarify if there are any pronounced differences. In this regard, the L-stellate dendritic fields look a lot like previous reconstructions of T-stellate cells, so how different are those 2 classes? How likely is it that previous studies claiming to record from T-stellate cells have actually mixed the populations?

Our data demonstrating carbachol-induced IPSCs onto both bushy and T-stellate cells is strong evidence for local inhibition provided by L-stellate cells. The only other interpretation is that inhibitory axons projecting from an unknown source external to CN reaches principal cells and happens to express presynaptic excitatory AChRs. Given the demonstrated excitatory actions of carbachol on L-cells (and not on other CN inhibitory neurons), we think Occam’s razor favors a local source of this inhibition. Indeed this is a powerful approach: unlike electrical stimulation of inhibitory sources in VCN and DCN in which the sources of inhibition are ambiguous, carbachol application allowed us to selectively activate L-stellate cells from all other known inhibitory cells in the CN. We agree that paired recordings would confirm the local connectivity. However, even after using numerous approaches suggested by the reviewer including cell-attached screening, puffing glutamate, potassium, cutting slices with a different orientation, establishing paired recording in the VCN remained extremely challenging. At this point, given the challenges of the COVID crisis, which of course shut down our lab and forced us to sacrifice a large fraction of our mouse colony, and given the extremely low yield of this experiment (we had stated we had only an N of 1 success in the paper), it would take an unreasonably long time to obtain such data. Another option as suggested by the reviewer, to perform a further detailed histological analysis, is very exciting but it is beyond the scope of the current work. We will consider this for a future research project when we are able to research again.

Inhibition from L-stellate cells may or may not be functionally redundant with the other classes of inhibitory cell. We showed that D-stellate cells and vertical cells do not respond to cholinergic agonist whereas many L-stellates showed increase in excitability. This suggests a specific role of cholinergic modulation of excitability of primary excitatory neurons through inhibitory neurons. We have expanded the Discussion of the possible function of cholinergic inputs onto L-stellate cells.

Stimulation of ANF without the DCN is an interesting thought, however as noted above, it could also activate collaterals of D-stellate cells. Prior to the lab’s shut down, we were actively pursuing a different targeted approach to activate only L-stellate cells in the CN.

However, we can report that just before the shutdown, we were able to pursue one of the reviewer’s suggested experiments, performing cell-fills followed by reconstruction of T-stellate cells. Our available data have been added in Figure 5—figure supplement 1, and show that L-cells may have more tightly ramified dendritic branches compared to T-stellate cells. We agree with the reviewer that, given the regular firing/chopper responses of L-cells, it is possible that previous papers describing T-stellate cells could reflect a mixed excitatory/inhibitory population.

3) Morphological characteristics of L-stellate cells: The axons of L-stellate cells are not clearly delineated in the experimental material. The drawings in Figure 5 show "axonal and dendritic arbors", but do not indicate which of the traced processes belong to which part of the cell. Although the data support the conclusion that these cells do not project contralaterally in any major way, it is unclear whether they might have other projections, such as to the DCN (it does not appear so from the reconstructions, but that is not a foregone conclusion as it is unclear how visible the axon was or whether it could be followed to its terminations). The "localness" of these cell's targets should be better qualified. In Figure 5, the heterogeneity of dendritic morphology among L-stellate cells is striking. Does dendritic morphology reflect electrophysiological differences?

The reviewer raised a valid point that applies to most anatomical reconstructions: we cannot prove that an axonal branch does not exist. Moreover, given the thin and profusely branched processes of L-cells, it was difficult to indicate dendrites or axons confidently in the majority of the reconstructions. However, we do feel our carbachol evidence clearly indicates that L-cells make local connections. Regarding the reviewer’s last point, we did not observe obvious correlation between anatomy and intrinsic properties.

4) Physiological properties of L-stellate cells: The demonstration of chopping response to direct current injection in Figure 7 might be expected for any regularly firing neuron. It would be more informative if the authors could demonstrate chopping in response to ANF stimulation. The diversity in input resistance and spike parameters across cells (e.g., in Figure 6) might have interesting functional consequences in these experiments and might be distinct from vertical cell responses. As the authors point out, it seems likely that there are at least two classes of L-stellate cells, but their different physiological characteristics have not been related to differences in morphology (Figure 5 and 6), so it is difficult to place them in a relevant context. It would be helpful if the authors could provide more information on this issue from their existing data set. Does carbachol sensitivity correlate with differences in AP shape parameters or topographic location within the VCN? In other words, does carbachol sensitivity reflect differences in cell type or simply the location of the cell with respect to cholinergic input fields?

It is true that a chopping response (which is a description of the neurons PSTH) is expected from regular firing neurons; indeed, it is surprising to us that this has not been explicitly demonstrated in CN before. We anticipated that spikes driven by synaptic stimuli should also show a chopping pattern, unless the EPSPs were extremely strong. Using data collected prior to shut down, we now illustrate chopping responses to ANF stimulation in Figure 7. We have recorded from L-stellate cells with ANF stimulation (n = 6). All the cells displayed chopping responses to ANF stimulation. Interestingly, the cells also show delayed firing even after the end of stimulation, which will be explored in a future study. We did not find any significant correlation between morphology and physiological characteristics. We also quantified the intrinsic properties of L-cells to carbachol sensitivity (n = 21 cells, 11 responded to carbachol and 10 with no carbachol response). The quantification is presented in Figure 6—figure supplement 1 and shows virtually no difference in spike properties of responders and non-responders.

[Editors' note: further revisions were suggested prior to acceptance, as described below.]

Additional quantitative information should be included, if possible, about the dendritic morphology of L- vs. T-stellate cells. In Figure 5—figure supplement 1: the analyses of L- vs. T-stellate cell dendritic morphology appears anecdotal, with just one L-stellate and 3 T-stellate cells reconstructed. This should be better documented, as this is a substantial part of the comparison between excitatory and inhibitory stellate cells in the VCN. It seems feasible even in a reduced research state to flesh out this data set enough to provide a statistical comparison.

We apologize for the near-anecdotal level of T-stellate data. We had been unable to do more drawings due to the unavailability of the cell tracing equipment during the lockdown. However, we have since been able to gain access and now provide more complete analysis, showing statistically different morphological features between all stellate subtypes.

Please provide a small orientation indicator for all figures with anatomical images. If all are in the same orientation (they appear to be), then this should be clearly stated somewhere in the manuscript. While I can figure out what the orientation is from clues in the sections, other readers could have difficulty.Figure 5: add statistical information for the added panel F legend, to be consistent with the rest of the legend.Figure 6—figure supplement 1: There is no reference to this information in the main text. Please add any statistical information to the figure legend.Introduction paragraph two: Lorente de No could not know neurons were "small inhibitory neurons", but he did describe many small cells in the VCN. Please reword (you could just remove "inhibitory" from the sentence).Introduction paragraph three: "receptive field" is still misleading here. They have a narrower dendritic field, from which a narrower receptive field would be inferred. Perhaps reword.Introduction paragraph three: "excitability of excitatory neurons " suggest -> "excitability of projection neurons "

All done

Results paragraph three: the link is incorrect: it is "http://mouse.brain-map.org/experiments/show/69874024".

Corrected

Subsection “Synaptic connectivity”: Just to reinforce this observation, delayed firing after trains of AN stimulation were also reported in a subset of "planar multipolar" cells by Xie and Manis, 2017. However, it is unclear whether the cells in that study correspond to the L-stellate cells in this study as the cell identification was not as robust.

Thanks for noting that.

Paragraph two in subsection “Synaptic connectivity”: number agreement: "cells have" (not "cells has")

Corrected

"For a few voltage-clamp experiments, we used an internal solution containing (in mM) 5 CsCl, 1 MgCl2, 4 Mg-ATP, 0.4 Tris-GTP, 5 EGTA, 14 Tris2-phosphocreatine, 4 Na2-ATP, 10 HEPES, 3 QX-314 (pH 7.2, 290 mOsm)." It is not clear to me how this electrode solution has an osmolarity of 290 mOsm. I still think there is something missing in this description. Please check.

Should be 115 CsCl. Fixed.